

# Carbonate chemistry in sediment pore waters of the Rhône River delta driven by early diagenesis (NW Mediterranean)

Jens Rassmann[1], Bruno Lansard[1], Lara Pozzato[1,2], Christophe Rabouille[1]

[1]Laboratoire des Sciences du Climat et de l'Environnement, LSCE/IPSL, CEA-CNRS-UVSQ-Université Paris Saclay, Gif-sur-Yvette, 91198, France

[2]Institut Méditerranéen d'Océanologie, CNRS-IRD-Université de Toulon-Aix Marseille, 13288 France

*Corresponding author*: Jens Rassmann (jens.rassmann@lsce.ipsl.fr)

**Abstract.** The Rhône River is the largest source of terrestrial organic and inorganic carbon for the Mediterranean Sea, and a large fraction thereof is buried or mineralized in the sediments close to the river mouth. The mineralization follows aerobic and anaerobic pathways with varying impacts on the carbonate chemistry in the sediment pore waters. This study focused on the production of dissolved inorganic carbon (DIC) and total alkalinity (TA) by early diagenesis at the sediment water-interface, consequential pH variations and the effect on calcium carbonate precipitation or dissolution. The sediment pore water chemistry was investigated during the DICASE cruise along a transect from the Rhône River outlet to the continental shelf. The concentrations of DIC, TA, $SO_4^{2-}$ and $Ca^{2+}$ were analyzed on bottom waters and extracted pore waters, whereas pH and oxygen concentrations were measured *in situ* using microelectrodes. The average oxygen penetration depth into the sediment was $1.7 \pm 0.4$ mm in the proximal domain and $8.2 \pm 2.6$ mm in the distal domain, indicating intense aerobic respiration rates. Diffusive oxygen fluxes through the sediment water interface range between 3 and 13 mmol $O_2$ m$^{-2}$ d$^{-1}$. The DIC and TA concentrations increased with depth in the sediment pore waters up to 48 mmol L$^{-1}$ near the river outlet and up to 7 mmol L$^{-1}$ on the shelf as a result of aerobic and anaerobic mineralization processes. Due to oxic processes, the pH decreased by 0.6 pH units in the oxic layer of the sediment accompanied by a decrease of the saturation state regarding calcium carbonate. In the anoxic part of the sediments, sulfate reduction was seen to be the dominant mineralization process and was associated to an increase of pore water saturation state regarding calcium carbonate. Ultimately anoxic mineralization of organic matter caused calcium carbonate precipitation as shown by large decrease in $Ca^{2+}$ concentration with depth in the sediment. The saturation state and carbonate



precipitation decreased in offshore direction, together with the carbon turnover and sulfate consumption in the sediments.

# 1 Introduction

The coastal ocean is a net sink of atmospheric $CO_2$ and plays an important role in the global carbon cycle (Hedges and Keil, 1995; Chen and Borgès, 2009; Bauer et al., 2013; Laruelle et al., 2013). It is not only a sink for atmospheric $CO_2$, but also a location where terrestrial organic and inorganic carbon is buried or recycled (Hedges and Keil, 1995; Cai, 2011). Due to strong pelagic-benthic coupling, a

large fraction of organic matter (OM) is mineralized in continental shelf sediments (McKee et al., 2004; Burdige, 2011; Bauer et al., 2013). Estuaries and delta regions are a very dynamic part of shelf regions characterized by a high carbon turnover (Hedges and Keil, 1995; Cai, 2011). They are the principal link between continents and oceans and receive inputs of terrestrial organic and inorganic carbon, in both, particulate and dissolved phases (McKee et al, 2004; Cai, 2011; Dai et al, 2012; Bauer

et al., 2013). An important fraction of these inputs remains on site and undergoes oxic and anoxic mineralization (Andersson et al., 2005; Aller and Blair; 2006; Chen et al., 2012). Despite their importance for the coastal carbon cycle, there is a lack of knowledge about the links between early diagenesis and the carbonate system in river dominated sediments (McKee et al., 2004). Aerobic and anaerobic reaction pathways contribute to the production of dissolved inorganic carbon (DIC), which

creates acidification of the bottom water and to the production of total alkalinity (TA), which increases the $CO_2$ buffer capacity of seawater (Thomas et al., 2009). Variations in DIC and TA affect the partial pressure of $CO_2$ ($pCO_2$) in seawater and ultimately the $CO_2$ exchange with the atmosphere (Emerson and Hedges, 2008). The processes by which TA is produced in the sediments are still not well understood : anaerobic respiration (denitrification, sulfate reduction, iron and manganese reduction)

seems to play a major role (Thomas et al., 2009, Krumins et al., 2013) but dissolution/precipitation of calcium carbonate can have a large impact on TA concentrations as well (Jahnke et al., 1997). Indeed, the changes in sediment pore water composition and pH can lead to over or under-saturation of calcium carbonate saturation state ($\Omega$) and therefore influence carbonate dissolution and burial in sediments (Mucci et al., 2000). Jahnke et al., (1997) showed using benthic flux measurements in deep sea




sediments and modeling tools, that TA fluxes across the sediment water interface (SWI) are related to metabolic carbonate dissolution. How much carbonates are affected by metabolic processes depends as well on the carbonate content in the sediments and the saturation state of the overlying waters (Jahke & Jahnke, 2004).

Using *in situ* microelectrode measurements, Komada et al. (1998) and Cai et al. (2000) investigated the
small scale changes in $pCO_2$, pH and DIC in marine sediments and the exchange fluxes associated. In continental shelf sediments, Mucci et al. (2000) showed that oxic mineralization can induce carbonate dissolution below the sediment-water interface and so extended the observations of Jahnke et al., (1997 and 2004) to coastal sediments. Burdige et al., (2008, 2010) showed that carbonate dissolution is also driven by oxic respiration in the shallow carbonated sediments of the Bahamas Bank. Concerning
anoxic processes, Van Capellen & Wang (1996) demonstrated that high manganese and iron contents in the sediments of the Skagerrak and associated OM mineralization can increase pore water pH by proton-consuming reduction processes of oxidized iron and manganese. They pointed at the complexity of the multiple competing reaction pathways in anoxic sediments and observed that the existing theoretical background (Froelich et al., 1979, Berner, 1980) was insufficient to disentangle them. In
regions with a high carbon turnover, sulfate reduction is a large contributor to anoxic early diagenesis and can even be the dominant mineralization process for OM (Mucci et al., 2000; Burdige and Komada, 2011; Pastor et al., 2011). Sulfate reduction slightly decreases pH (Jourabchi et al, 2005; Soetaert et al., 2007), but nevertheless, it tends to enhance carbonate precipitation because of its coupling with precipitation of sulfide minerals from iron oxides (Gaillard et al., 1989; Mucci et al., 2000; Burdige,
2011). As an example, in sapropelic sediments from a Mangrove Lake, Mackenzie et al. (1995) reported a stable pH throughout the sulfate-reduction zone and a buildup of supersaturation with respect to carbonate with depth. These results contrast with a theoretical point of view where sulfate reduction was supposed to lead to carbonate dissolution because of the pH decrease (Jourabchi et al., 2005). Even today, the reproduction of measured pore water profiles in the sediments and the estimation of TA and
DIC fluxes across the SWI by modeling is very challenging (Arndt et al, 2013; Krumins et al., 2013; Jourabchi et al., 2005). In addition, the magnitude of DIC and TA fluxes across the SWI are not well constrained and can show important variations between different study sides (Mucci et al., 2000).



In order to improve our understanding of the influence of early diagenesis of organic matter on carbonate dissolution/precipitation, we designed a study in the Rhône River delta in the Mediterranean Sea, which displays a range of biogeochemical characteristics (Lansard et al., 2009; Cathalot et al., 2010; Cathalot et al. 2013). Indeed, the Rhône River delta receives inputs of terrestrial organic and inorganic carbon, in both particulate and dissolved phases which decrease with the distance to the river mouth. An important fraction of these inputs remains on site and undergoes mineralization in the sediments (Pastor et al., 2011a). Therefore sediments display strong spatial gradients in biogeochemical parameters such as nutrients, organic and inorganic carbon, affecting the diagenetic transport-reaction network (Bourgeois et al., 2011; Lansard et al., 2008). High sedimentation rates and resuspension events make this environment very dynamic and heterogeneous (Cathalot et al. 2010). In extreme cases near the river outlet, the downward advection due to high sedimentation rates can compete with diffusive transport of dissolved species like DIC and TA. We investigated a transect of stations characterized by various biogeochemical conditions (from oxic-dominated to sulfate reduction-dominated sediments). We used a combination of *in situ* oxygen and pH microelectrode measurements and pore water analysis of DIC, TA, $SO_4^{2-}$ and $Ca^{2+}$ concentrations to cover different vertical scales. We calculated and discussed the calcium carbonate saturation state in regard to the different intensity of biogeochemical processes in these river-dominated sediments.

## 2 Study site and methods

### 2.1 The Rhône River delta

With a drainage basin of 97 800 km$^2$ and a mean water-discharge of 1700 m$^3$ s$^{-1}$, the Rhône River is the largest river of the Mediterranean Sea in terms of fresh water discharge, inputs of sediment and terrestrial organic and inorganic matter (Pont, 1997; Durrieu de Madron et al., 2000; Sempéré et al., 2000). The Rhône River mouth is a wave-dominated delta located in the microtidal Mediterranean environment of the Gulf of Lions (Sempéré et al., 2000). Its river plume is mostly oriented southwestward, due to the Coriolis Effect and the wind forcing (Estournel et al., 1997). The annual discharge of particulate inorganic carbon (PIC) is estimated to 0.68 ± 0.45 10$^9$ gC (Sempéré et al.,



2009). The total particulate organic carbon (POC) deposition in the Rhône delta system (265 km$^2$) is about $100 \pm 31$ $10^9$ gC y$^{-1}$ where the deltaic front accounts for nearly 60 % of the total POC deposition (Lansard et al., 2009). Off the river mouth, the deposited sediments are of cohesive nature and composed of fine grained sediments with more than 90 % of silts and clays (Roussiez et al., 2005; Lansard et al., 2007). Previous studies have shown that the carbonate content in the surface sediments varies between 28 and 38 % (Roussiez et al., 2006) and the content of OC between 1 and 2 % (Roussiez et al, 2005, 2006; Lansard et al., 2008, 2009). The PIC in the sediments is composed by autochtonous and allochtonous carbonates. The most aboundant calcifying organisms in this area are foraminifera (Mojtahid et al., 2010).

The seafloor bathymetry shows that the delta is divided in three zones, characterized by different water depth, sedimentation rate and strength of continental slope. Got and Aloisi, 1990, defined three major domaines that we call : Proximal domain, in a radius of 2 km from the river outlet with water depth ranging from 10 to 30 m, Prodelta domain, between 2 and 5 km from the river mouth with depth ranging from 30 to 70 m and Distal domain, with depth between 70 and 80 m passed the 5 km from the river mouth. Annual sedimentation rates reach up to 30-48 cm yr$^{-1}$ close to the river mouth (Charmasson et al., 1998) and rapidly decrease below 0.1 cm yr$^{-1}$ on the continental shelf (Miralles et al. 2005). The sea floor in this region is a dynamic environment with important heterogeneity concerning diagenetic activities, sediment pore water profiles and exchange fluxes at the sediment-water interface (Lansard et al., 2009; Cathalot et al., 2010).

Diffusive oxygen fluxes into the sediment show spatial variability, both with the distance from the river mouth (decreasing in offshore direction) and on the horizontal scale of a few cm² (Lansard et al., 2009; Pastor et al., 2011b). Anoxic mineralization processes play a major role in the Prodelta sediments and are dominated by iron and sulfur cycling (Pastor et al., 2011a).

## 2.2 The DICASE cruise

The DICASE oceanographic cruise took place in the Gulf of Lions from the 2$^{nd}$ to the 11$^{th}$ of June 2014 on board of the RV Tethys II (http://dx.doi.org/10.17600/14007100). Ten stations have been sampled



along the main direction of the Rhône River plume. Their positions and main characteristics are shown
in Figure 1 and in Table 1. The stations were chosen between 2 and 25 km distance from the Rhône
River mouth, covering a bathymetric gradient ranging from 20 m to 80 m of water depth and
representing the three different domaines (A,Z : Proximal Domain; AK, B, K, L : Prodelta Domain and
C,D,E,F : Distal Domain). The stations in the proximal domain A and Z have been sampled twice, in
order to investigate spatial  variability at these two stations. During this cruise, a benthic lander was
used to measure in situ oxygen and pH micro profiles and sediment cores were taken for pore water
extraction and solid phase analysis.

## 2.3 In situ measurements

To measure *in situ* oxygen and pH micro profiles at the sediment-water interface, an autonomous
lander (Unisense®) was used. This lander is equipped with a high precision motor capable to move
simultaneously five oxygen microelectrodes (Revsbech, 1989), two pH microelectrodes and a resistivity
probe (Andrews and Bennet, 1981) with a vertical resolution of 100 μm. The recorded oxygen profiles
were calibrated using oxygen concentrations measured in bottom waters (BW) by Winkler Titration
(Grasshoff et al., 1983) and the zero oxygen measured in the anoxic zone (Cai and Sayles, 1996). The
location of the SWI was positioned where the strongest vertical oxygen gradient was situated (Rabouille
et al., 2003). The calibration of the pH electrodes was carried out using NBS buffers, thus allowing the
estimation of the slope of the pH variation at onboard temperature. The slope was then recalculated at in
situ temperature and the electrode signal variation was transformed into pH changes. The pH of bottom
waters was determined using the spectrophotometric method with m-cresol purple following (Clayton
and Byrne, 1993; Dickson et al., 2007)  and pore water pH on the total proton scale ($pH_t$) was
recalculated using this BW value the micro electrod measured pH variations. At each depth, the profiler
was waiting for 20 seconds to stabilize the electrode before measurements were recorded. Each data
point is an average of five measurements carried out at every depth. For all *in situ* profiles, the signal
drift of each microelectrode was checked to be inferior to 5 % between the beginning and the end of the
measurements. The slope of the pH electrodes was checked to be at least 95 % of the theoretical slope



from the Nernst equation of -59 mV per pH-unit at 25 °C. At each station, 5 oxygen profiles and two pH profiles were measured simultaneously on a surface of 109 cm².

## 2.4 Calculation of oxygen fluxes across the sediment-water interface

Sediment oxygen uptake rate has been widely used to assess benthic OC mineralization during early diagenesis. Total oxygen uptake (TOU) rate can be split into two parts: (i) oxygen uptake rate of diffusive nature (DOU), and (ii) advective oxygen uptake. The DOU rates across the SWI were calculated using Fick's first law (Berner, 1980):

$$DOU = -D_s \cdot \varphi \frac{d[O_2]}{dz}\bigg| z = 0 \quad (1)$$

with:

$D_s$ : apparent diffusion coefficient adjusted for diffusion in porous environment calculated following $D_s = \frac{D_0}{1+3 \cdot (1-\varphi)}$ where $D_0$ is the diffusion coefficient in free water according to (Broecker and Peng, 1974)

$\phi$ : sediment porosity

$\frac{d[O_2]}{dz}\bigg| z = 0$ : Oxygen gradient at the sediment-water interface

## 2.5 Sampling and *ex situ* measurements

Bottom water samples were collected with a 12-L Niskin bottle as close as possible to the sea floor at each station. On these samples, temperature was measured using a digital thermometer with a precision of 0.1 °C and salinity was measured with a salinometer with a precision of 0.1. pH and concentrations of DIC, TA and oxygen were measured on board as soon as possible within one hour for pH and within six hours for DIC and TA. The pH of seawater was measured using a spectrophotometer and m-cresol purple as dye (Clayton and Byrne, 1993; Dickson et al., 2007) with uncertainties smaller than 0.01 pH units. Oxygen concentrations were determined using Winkler titration with an average uncertainty of



0.4 µmol L$^{-1}$. All DIC concentrations (bottom waters and pore waters) were measured on a DIC analyzer (Apollo SciTech$^{®}$) using 1 ml sample volume with 4 to 6 replicates. The principle of the method is to acidify the sample with phosphoric acid of 10 % concentration to transform all forms of DIC into $CO_2$. The sample is then outgassed using ultra-pure nitrogen as vector gas. The degassed $CO_2$ is quantified by a LICOR containing a non-dispersive infrared detector (NDIR). To calibrate the method, certified reference material (CRM-batch #122, provided by A. Dickson, Scripps Institution of Oceanography) was used at least twice a day to confirm accuracy of the measurements. TA concentrations were measured in a potentiometric open cell titration on 3 ml sample volume (Dickson et al., 2007). Uncertainties of DIC and TA measurements in the sediment pore waters were below 0.5 %.

Sediment cores were sampled using a UWITEC$^{®}$ single corer. After sampling, the cores were rapidly introduced in a glove bag under $N_2$ atmosphere to avoid oxidation and pore waters were extracted using Rhizons with pore size of 0.1-0.2 µm (Seeberg-Elverfeldt et al., 2005). The Rhizons had been degassed and stored in a $N_2$-filled gas tight box before use. Pore waters were typically extracted with a 2 cm vertical resolution and split into subsamples for DIC, TA, $SO_4^{2-}$ and $Ca^{2+}$ analysis. Sulfate concentrations were measured in the laboratory using a turbidimetric method (Tabatai, 1974). Concentrations of calcium ions were measured using ICP-AES (Ultima 2, Horiba$^{®}$) by the "Pôle Spectrométrie Océan" in Brest (France) with a relative uncertainty of 0.75 %. The calcium concentrations were salinity corrected by assuming constant $Na^+$ concentrations with depth in the pore waters, in order to avoid any evaporation effects due to the sample storage.

At each station, additional cores were taken for solid phase analysis. To establish porosity profiles, fresh sediment samples were weighted, dehydrated during one week at 60 °C and weighted again. Knowing the salinity and density of seawater and sediment, porosity was calculated from the weight loss after drying. Total carbonate content of the solid phase was analyzed using a manocalcimeter with uncertainties of 2.5 % of $CaCO_3$. A manocalcimeter consists in a small, gastight container where the sediment can be acidified with HCl to dissolve calcium carbonates. The resulting increase of pressure is measured with a manometer and is directly proportional to the carbonate content of the sediment sample. Sediment samples have also been analyzed via X-Ray diffraction on a X-Pert Pro



diffractometer, using the θ-θ-technique with the K-α-line of copper, to quantify the calcite/aragonite

proportion. The uncertainties of the XRD measurements were below 5 % of the aragonite proportion

(Nouet and Bassinot, 2007).

## 2.6 Calculation of carbonate speciation, CaCO$_3$ saturation states and pH in pore waters

According to Orr et al. (2015), the best way to compute the 12 parameters of the carbonate system at *in*

*situ* conditions is to start with DIC and TA concentrations. The thermodynamic constants proposed by

Lueker et al. (2000) were used to calculate data about DIC speciation and pore water pH by the program

CO2SYS (Lewis and Wallace, 1998). The calcium carbonate saturation state is expressed as the

solubility product of calcium and carbonate ions concentrations divided by their solubility constant k$_{sp}$:

$$\Omega_{Ca} = \frac{[Ca^{2+}][CO_3^{2-}]}{k_{sp}} \quad (2)$$

The solubility constant k$_{sp}$ was calculated for *in situ* temperature, salinity and pressure following

(Millero et al.,1979;  Mucci, 1983; Millero, 1995). The existing numerical tools are developed for the

water column, but we used them in the sediments knowing that pore water concentrations (DIC, TA,

nutrients) are much larger than those in the water column. Despite this potential artifact, the calculated

outputs (e.g. pH) agree with our measurements.

## 2.7 Principal diagenetic reactions and their influence on the carbonate system

Table 2 summarizes the main diagenetic reactions (simplified) and their impact on the DIC and TA

concentrations. The dissolution and dissociation of CO$_2$ in seawater leads to the formation of carbonic

acid (R1), the consumption of CO$_3^{2-}$ and ultimately leads to carbonate dissolution (R2). DIC is always

produced by OM mineralization, whereas the TA budget of these reactions and the resulting pH

variation can be positive or negative. Aerobic mineralization leads to a decrease of pH without TA

production (R3) and finally decreases Ω. In the sediments, oxygen is also used to reoxidise reduced

species, a process that decreases pH even more strongly than aerobic respiration (R4-6) and thus



reoxidation decreases Ω as well. In contrast, anaerobic mineralization causes much weaker pH drops compared to the oxic processes and can even increase pH (R7-9). The precipitation of sulfur minerals does not affect the amount of pore water DIC, but can have an important influence on pH and TA (R10-13). The two reactions R14 and R15 deal with the coupling of sulfate reduction and methanogenesis and its impact on DIC.

# 3 Results

## 3.1 Bottom waters

In June 2014, the Rhône River water level was low and close to 1000 $m^3 s^{-1}$ since the previous 2 months. Therefore, the spread and thickness of the Rhône River plume were very limited and bottom waters were not influenced by the river outflow, even close to the river mouth. Bottom water temperature, salinity, $O_2$, DIC and TA concentrations, pH, $SO_4^{2-}$ concentrations and bottom water $pCO_2$ are given in Table 1. Salinity remained very constant close to the sea floor, whereas temperature decreased with water depth from 16.8 to 14.3°C. Bottom waters were well oxygenated and oxygen concentrations decreased also with increasing water depth. Whereas DIC and TA concentrations varied slightly and the TA/DIC ratio in the bottom waters of all stations was $1.1 \pm 0.02$. The pH of bottom water showed some local variability with a general decrease in offshore direction. $SO_4^{2-}$ concentrations were constant between the stations and showed typical values for seawater around 30mmol $L^{-1}$. $pCO_2$ showed oversaturation compared to the atmosphere at all stations.

## 3.2 The oxic layer

Figure 2 shows all oxygen profiles measured *in situ* during the DICASE cruise. In the proximal domain, the oxygen penetratin depth (OPD) measures $1.7 \pm 0.4$ mm into the sediment, $3.3 \pm 1.3$ mm in the prodelta domain and $8.2 \pm 2.6$ mm in the distal domain. Some profiles show the presence of burrows creating small oxygen peaks below the oxygen penetration depth. The diffusive oxygen uptake rate (DOU) calculated from the measured oxygen profiles are shown on Figure 3 as a function of the distance to the river mouth in the direction of the river plume. The positive value significates an uptake



of $O_2$ into the sediment. DOU decreases exponentially with the distance from $12.3 \pm 1.1$ mmol L$^{-1}$ m$^{-2}$ d$^{-1}$ at station A towards the minimum flux of $3.8 \pm 0.9$ mmol L$^{-1}$ m$^{-2}$ d$^{-1}$ at station F.

*In situ* pH micro profiles were measured in the top 4 cm of the sediment at all stations (Fig. 4). Immediately below the sediment-water interface, the pH drops about 0.6 to 0.7 units in the oxic layer. Similarly to the oxygen micro profiles, the pH gradient in the OPD is stronger close to the river mouth and weaker in the distal domain. Just below the first drop, pH increases for 0.1-0.2 pH units and tends towards an asymptotic value between 7.4 to 7.6. The pH inflexion point, i.e. where the decrease stops and pH starts increasing, is located deeper in the distal zone than in the proximal zone, just below the OPD. The pH profiles show very high heterogeneity, even at one station.

## 3.3 DIC and TA pore water concentrations and calculated pH

Figure 5 shows the DIC and TA pore water profiles measured during the DICASE cruise. All pore water gradients across the sediment-water interface were strongest close to the river mouth and decreased in offshore direction. At the SWI at all stations, the DIC gradients were stronger than the TA gradients. Despite spatial heterogeneity in the sediments, the three major areas defined by!

Got and Aloisi, (1990) in this region appear clearly with different biogeochemical gradients. Stations of each group will be discussed separately. In the proximal domain (stations A and Z), DIC and TA concentrations increase immediately below the SWI and reach a maximum value of 48 mmol L$^{-1}$ at 20 cm depth in the sediments where the concentrations seem to stabilize. In the prodelta domain (stations AK, B, K and L), DIC and TA concentrations increase to values of 5 mmol L$^{-1}$ in the first 10 to 15 cm depth. Below this depth, the gradients become stronger and TA and DIC concentrations increase up to 12 to 15 mmol L$^{-1}$ at the bottom of the cores, i.e. around 25 cm depth. This succession of two different gradient shapes in the TA and DIC profile is also observed in the distal domain (stations C, D, E and F), but the absolute values of the gradients are weaker. In the first 10 to 15 cm, the concentrations reach values of 3.5 mmol L$^{-1}$ to increase up to 5 to 7.5 mmol L$^{-1}$ at the bottom of the core. These very high DIC concentrations in the sediment are related to large DIC and TA gradients which are 4 to 10 times stronger in the proximal domain than at the other sites. The DIC and TA pore water profiles are well



correlated in each core and the concentrations show a linear correlation with a determination coefficient of $r^2 > 0.99$ (130 data points). The slope of the linear correlation decreases from 1.05 to 0.90 when moving seawards.

Figure 6 shows pH profiles in the sediments that were calculated from TA and DIC concentrations using CO2SYS. The pH is reported on the total proton scale. In the first mm, the pH drops at all stations
due to aerobic respiration. Below the oxygen penetration depth, pH varies between 7.2 and 7.8 and converges towards the range of 7.4 to 7.6. The calculated values of pH overlay with the values measured by the pH-microelectrodes.

## 3.4 Calcium and sulfate concentrations

The calcium pore water profiles are shown in Fig. 6. At all stations, bottom water $Ca^{2+}$ concentration
varies between 10 and 11 mmol $L^{-1}$. In the proximal domain, the $Ca^{2+}$ concentration decreases just below the SWI to reach a minimum of 2 mmol $L^{-1}$ at 15-20 cm depth, where DIC and TA concentrations reach a maximum and sulfate concentration a minimum. In the prodelta domain, the $Ca^{2+}$ concentration remains stable with the depth until 10-15 cm depth related to the weaker TA and DIC gradients (Fig. 6). Below this depth, where the TA and DIC gradients increase, $Ca^{2+}$ decreases to values
around 7 mmol $L^{-1}$ at the bottom of the cores. The distal domain is characterized by constant $Ca^{2+}$ concentrations which remain above 10 mmol $L^{-1}$.

In extracted sediment pore water, sulfate concentrations range from 5 to 32 mmol $L^{-1}$ from the surface down to 30 cm depth. Our measurements indicate strong sulfate consumption rates in the proximal domain (Fig. 8) where DIC and TA gradients are strong as well. Even in the first cm below the SWI in
the proximal domain sulfate concentration decrease compared to the bottom water. In the prodelta domain, sulfate reduction starts to occur between 10 and 15 cm depth (Fig. 8), the same depth where TA and DIC gradients increase. In the distal domain no significant sulfate reduction seems to occur in the first 30 cm, as sulfate concentration remains constant (Fig. 8) and TA and DIC gradients are low compared to the other domains.




## 3.5 Solid carbonates

The carbonate content of the solid phase scattered around 35 % at all stations, from the surface down to 30 cm. The composition of sedimentary $CaCO_3$ was dominated by calcite (> 95 %), with a small fraction of magnesian calcite (> 5 %), and less than 2 % of aragonite (data not shown). Taking into account the precision of the DRX measurements of ± 5 %, we cannot differentiate if both these phases were present in the sediments of the study area or if it is a measurement error.

## 3.6 Calcium carbonate saturation state

In this study, we only report on the $\Omega_{calcite}$ since calcite is dominant and aragonite is insignificant in the sedimentary $CaCO_3$. The results for the calcite saturation state in pore waters are shown on Fig. 8. The saturation state drops in the oxic layer. In the proximal domain (Fig. 8), the saturation state increases immediately below this first drop to reach very high values of around 5 to 10. In the prodelta domain (Fig. 8), the saturation state remains very close to 1 at a depth between 5 and 10cm before increasing to super saturation (3 to 4) below 10 to 15 cm depth. In the distal domain (Fig. 8), the saturation state shows no variation below the first drop.

## 4 Discussion

### 4.1 The impact of oxic and suboxic processes on the carbonate system

The upper part of the sediment, is defined as the oxic zone, supporting aerobic respiration (R3). Generally, the oxygen penetration depth (OPD) is related to aerobic respiration rates (Cai and Sayles, 1996). Aerobic respiration consumes $O_2$ to mineralize organic matter, produces metabolic $CO_2$ in the sediment pore water, increases the DIC concentration, lowers pH and possibly decreases the $CaCO_3$ saturation state (R1 and Cai et al., 1993, 1995). The OPD and oxygen fluxes are therefore key parameters to assess the effect of aerobic respiration on calcium carbonate in the sediment (Jahnke et al., 1997; Jahnke and Jahnke, 2004).



In the Rhône River delta, the OPD increases with water depth and distance from the Rhône River
mouth. Very similar in situ OPD were reported for the sediment of the same study area in previous
studies (Lansard et al., 2008, 2009; Cathalot et al., 2013). These low values of $O_2$ penetration depths are
classical for river-dominated ocean margins and they depend mainly on the sedimentation rate, the flux,
the age and the oxidation state of OM (Lansard et al., 2009, Cathalot et al. 2013). Few in situ $O_2$ profiles
show oxygen peaks at depth below the OPD. These are likely the effect of sediment bioturbation by the
benthic macrofauna. As reported by Bonifácio et al., (2014), the macrofauna community is dominated
by polychaetes and the highest activity is found in the prodelta domain. Nevertheless, comparisons
between TOU and DOU rates have demonstrated that DOU account for about 80 % of total oxygen
uptake rate into the sediments (Lansard et al., 2008). As a consequence, diffusive transport is dominant
compared to advective transport and bioturbation (i.e. bioirrigation and bioventilation). Diffusive $O_2$
fluxes calculated from in situ 1D micro profiles (Fig. 2) are therefore representative for total oxygen
uptake rates. As shown on Fig. 3, the diffusive oxygen fluxes into the sediment decrease exponentially
with the distance from the river mouth, from $12.3 \pm 1.1$ mmol $O_2$ m$^{-2}$ d$^{-1}$, close to the Rhône River
mouth, to $3.8 \pm 0.9$ mmol $O_2$ m$^{-2}$ d$^{-1}$ offshore. Despite spatial and temporal variability, similar oxygen
fluxes have been reported by previous studies in the same area (Lansard et al., 2008, 2009; Cathalot et
al., 2010). According to Pastor et al., (2011a), the POC flux in the proximal domain is one order of
magnitude higher than in the offshore regions of the Rhône prodelta. Following model estimates, this
OM flux and especially fast fraction supports oxygen consumption as it is completely mineralized in the
oxic layer (Pastor et al., 2011a).

During aerobic respiration, the ratio of oxygen used to mineralize OM is close to 1, conforming to the
stoichiometry of equation (R3). As a result, DIC concentrations increase just below the SWI at all
stations (Fig. 5). The balance between $O_2$ flux and carbon oxidation in the sediment is affected by $O_2$-
consumption linked to the oxidation of inorganic species produced via anoxic organic carbon
degradation ($NH_4^+$, $Fe^{2+}$, $Mn^{2+}$ and $HS^-$). The oxidation of reduced diagenetic products has a profound
effect on pore water $O_2$ and pH profiles in $O_2$ limited sediments (Cai and Reimers, 1993). These
reactions (R4 to R6), in addition to aerobic bacterial respiration, consume TA and affect porewater pH
and therefore the calcium carbonate saturation state. There is a large contribution of anoxic processes to



total OM mineralization in sediments near the Rhône River mouth, certainly due to large inputs of fresh organic material combined with high sedimentation rates (Pastor et al., 2011a). The diagenetic by-products originally produced during anoxic organic matter mineralization are almost entirely

precipitated (> 97 %) and buried in the sediment, which leads to a relatively low contribution of the re-oxidation of reduced products to total oxygen consumption. Still, about 10 to 40 % of the oxygen flux is used to oxidize reduced species of iron and manganese, contributing to lower pH (Pastor et al., 2011a). Again, the upward flux of reduced species in the sediments is higher in the proximal domain than in the others. Offshore, less OM is available and the diagenetic activity is weaker, providing less reduced

species from deeper sediment layers.  pH drops below the SWI, caused by all oxic processes, are visible on the in situ pH micro profiles  and decreases until the OPD is reached (Fig. 2 and 4). As the OPD are smaller and the oxygen fluxes are higher in the proximal domain, the pH minimums are reached at shallower depth in the sediment than in the other domains. The pH drop is lowering $\Omega$ by consuming carbonate ions (Emerson and Hedges, 2008; Jourabchi et al., 2005). The decrease of $\Omega$, due to both

aerobic respiration and the oxidation of reduced species, is clearly visible between the first two points located above and below the SWI interface (Fig. 8).

Just below the oxic layer, OM mineralization via $MnO_2$ and $Fe(OH)_3$ reduction (R7-8) increases pH and releases large amounts of TA. The first pore water data point sampled in the sediments represents a mixture of oxic and anoxic pore water. Therefore, we potentially over estimate $\Omega$ in the oxic layer based

on calculations from pore water concentrations (Cai et al., 2010). Different measurements in the deep sea revealed that $\Omega$ shows a minimum in the oxic layer (Cai et al., 1993, 1995, 1996; Hales and Emerson, 1997; ). As pH decreased at all stations to the same value, but the TA and DIC gradients at the interface are the strongest in the proximal domain, $\Omega$ should show the highest values in the oxic sediments of the proximal domain and decrease in offshore direction. High TA concentrations in the

oxic layer resulting from anoxic OM mineralization below, prevent the carbonate saturation state from getting undersaturated. Therefore potential dissolution in the oxic layer would most likely occur in the distal domain, but could be inhibited in the proximal domain.



In agreement with current understanding of anoxic diagenesis, the observed pH increase of 0.1 to 0.2 units below the OPD can be attributed to OM mineralization via reduction onf iron and manganese (R7
and R8). These anoxic reactions release TA and increase pH in the oxic-anoxic transition zone (Aguilera et al., 2005; Jourabchi et al., 2005). This pH increase and the release of important quantities of TA create an important increase in the pore water saturation state ($\Omega$). Previous works showed that the turnover of Fe and Mn is important in the sediments close to the river mouth (Pastor et al., 2011a).

## 4.2 Sulfate reduction and its impact on carbonate chemistry

With sulfate concentration in seawater around 30 mmol L$^{-1}$, $SO_4^{2-}$-reduction can generate large amounts of DIC and TA during organic matter mineralization through sulfate reduction. Indeed, in organic rich sediments, sulfate reduction can account for the majority of OM mineralization (Gaillard et al., 1989; Jourabchi et al., 2005; Burdige, 2011; Fenschel et al., 2012) (R9). Following equation R9, two units of DIC and TA are produced for one unit of sulfate consumed (Mucci et al., 2000; Krumins et al., 2013).
Fig. 9 shows the diffusion corrected $\Delta DIC/\Delta SO_4^{2-}$ ratio in the pore waters of the proximal domain. This ratio compares the difference of pore water concentration of sulfate or DIC at a given depth with the concentration in the bottom water at the same station corrected for molecular diffusion following the equation proposed by (Berner, 1980) and using the diffusion coefficients determined by (Li and Gregory, 1973). Below 10 cm depth, the observed $\Delta DIC/\Delta SO_4^{2-}$ ratio of 1.9 ± 0.3 is statistically similar
to 2 which indicates that sulfate reduction is dominant below this depth. The large standard deviation observed around the mean can be linked to higher oxidation states of organic matter which lowers the $SO_4^{2-}$ requirement for mineralization, carbonate precipitation lowering DIC concentrations or a coupling of sulfate consumption with methane through the anaerobic oxidation of methane what increases DIC concentrations (Burdige and Komada, 2011; Antler et al., 2014). These possibilities may be acting in
the Rhone Delta proximal zone, as a large fraction of the OM mineralized in the proximal domain is of terrestrial origin, aged and already partly oxidized before being deposited on site (Cathalot et al., 2013), as calcium carbonates precipitate (Fig. 7) and as the presence of methane has been reported by Garcia-Garcia et al., (2006).




As shown by (Burdige, 2011, Burdige and Komada, 2011), the interaction of all diagenetic pathways
are hard to disentangle and do not provide clear evidence of changes in $\Delta DIC/\Delta SO_4^{2-}$ ratio. Nonetheless,
the value of the observed $\Delta DIC/\Delta SO_4^{2-}$ ratio (1.9 ± 0.3) points towards the dominance of sulfate
reduction in the deeper layers of the sediment (below 10 cm depth). Sulfate reduction is also attested by
the co-production of alkalinity and DIC (Fig. 5) and according to (Krumins et al., 2013), by far the most
important alkalinity producer in marine sediments. Sulfate reduction creates a TA/DIC ratio very close
to 1 in the pore waters of the proximal zone sediments. This situation is very similar to Mangrove Lake
sediments (Mackenzie et al., 1995) where depletion of sulfate is almost complete and DIC and TA
concentrations build up to 40 mmol $L^{-1}$ in the sediment pore waters, or to other coastal environments
(Burdige, 2011; Antler et al., 2014). No other reaction in the anoxic zone has a TA/DIC production ratio
near 1. As pH is buffered, probably by precipitation of FeS and $FeS_2$ (R12), this large increase of
alkalinity is accompanied, in the proximal zone, by a large increase of the saturation state of pore waters
with respect to calcite (Fig. 8) up to values of oversaturation ($\Omega$) from 5 to 10. The effect of sulfate
reduction and the carbonate saturation state has been a matter of debate since the early work of Ben-
Yaakov (1973). Indeed, sulfate reduction produces large quantities of both alkalinity which increases $\Omega$
and protons which decrease $\Omega$. This has been summarized in Jourabchi et al.'s model (2005) by
estimating that sulfate reduction would lead to decrease of $\Omega$ if it was the only ongoing reaction. The
sediments from the proximal area of the Rhône Delta show, on the contrary, that pH stabilizes between
7.2 an 7.6 driven by sulfate reduction which generates an increase of saturation state with respect to
calcite correlated with sulfate decrease (Fig. 8). This situation is very similar to (Mackenzie et al., 1995)
and (Mucci et al., 2000) who also showed an increase of $\Omega$ when sulfate reduction is significant. Using a
closed system model, (Ben Yaakov, 1973) estimated that oxidation of $HS^-$ coupled to iron hydroxide
reduction with FeS precipitation (as in R11 or R12) would buffer or even increase pH. Charles et al.,
(2014) suggested, that OM mineralization in the prodelta of the Rhône could be coupled to pyritisation.
The Rhône River is known to be the most important riverine input of iron into the Mediterranean Sea
(Guieu et al., 1991) with an iron content varying between 2 and 4 % in the solid phase discharge. In the
proximal zone of the Rhone Delta, dissolved sulphide is absent from the first tens of centimeters in the
sediment (Pastor et al., 2011a) which indicates that re-oxidation and/or precipitation of sulphide is



occurring in these sediments. Pastor et al, (2011a) estimated that 97 % of the reduced species from the anoxic sediments precipitate before diffusing to the oxic layer and that sulfides are the limiting factor for pyrite precipitation in this environment. With this FeS coupling, pH is stabilized or tends even to

increase and a large oversaturation with respect to calcium carbonâte is created due to produced carbonate ions.

In the proximal domain, the large supersaturation with respect to calcite, induces calcite precipitation as evidenced by a large decrease of dissolved calcium in the pore waters (Gaillard et al., 1989; Boudreau et al., 1992). Indeed, $Ca^{2+}$ concentration decreases by 9 mmol $L^{-1}$ between the bottom water and 25 cm

depth in proximal sediments. In the prodelta domain (Fig. 7), a similar set of reaction involving sulfate reduction and sulphide re-oxidation and precipitation is also visible with lower amplitude as sulfate depletion is only 5 mmol $L^{-1}$. Oversaturation with respect to calcite reaches values ranging from 3-4 only below 15 cm, and the $Ca^{2+}$ decrease is limited and arises deeper. In the distal zone where $\Omega$ is around 2 down to 25 cm, no calcium decrease is visible indicating that precipitation does not occur.

The large precipitation of calcium carbonate in the proximal zone may have implication on the $CO_2$ source from the sediment. Indeed, calcium carbonate precipitation generates $CO_2$ (R2) which can then be exported to the water column. In addition, calcium carbonate precipitation consumes TA. Thus pH and $\Omega$ are lowered in the bottom waters by these anoxic processes and lead estuarine environments to a high $pCO_2$.

As the majority of the reduced species is precipitated in the anoxic layer so they do not contribute to lower pH in the oxic layer and as the produced alkalinity fluxes are high, calcium carbonates could even be preserved in the oxic layer. Therefore, the alkalinity build up below could diffuse across the oxic sediment layer and contribute to buffer bottom waters and to increase $CO_2$ storage capacity. Without this TA flux, the $pCO_2$ of the bottom waters in the prodelta of the Rhône would be much higher than

observed.





# 5 Conclusions

The results of this work indicate the existence of three major domains in the Rhône prodelta
characterized by different degres of organic and inorganic particulate carbon interactions. Close to the
river mouth where the carbon turnover is the most important, the biogeochemical gradients are the
strongest resulting in high chemical fluxes across the SWI. This confirms that the biogeochemistry in
the prodelta region is driven by the import and degradation of terrestrial organic matter.

The oxic reactions produce $CO_2$ and create a pH drop of 0.6 to 0.8 pH units and reduce $\Omega$. As a
consequence calcium carbonate may dissolve in the oxic layer, but dissolution could not be put to
evidence in this study. The majority of oxygen is used for OM mineralization as most of reduced
species precipitate in anoxic sediments and does not contribute to oxygen consumption. The
mineralization of OM by Fe and Mn oxides increases pH and $\Omega$ just below the oxic layer in several mm
depth.

The strong TA and DIC gradients observed in the sediments of the Rhône prodelta suggest that OM
mineralization is dominated by anaerobic processes. Close to the river mouth, where the organic carbon
content in the sediments is highest, sulfate reduction is the dominant mineralization process for OM
degradation creating a strong coupling between TA and DIC pore water profiles. Despite its theoretical
lowering effect on pH, sulfate reduction is related to an increase of $\Omega$ by important alkalinity production
and via the simultaneous pH increase by precipitation of iron-sulfate-minerals. As a result, pore waters
are over saturated regarding calcite at all sampled stations. Calcium carbonate precipitation occurs in
the proximal and in the prodelta domain, depleting the majority of dissolved calcium ions in the
proximal domain. This carbonate precipitation represents an additional $CO_2$ source from the sediments
to the water column. But due to important anoxic TA production, the $pCO_2$ of bottom waters stays
relativly low compared to the important release of DIC due to OM mineralization.




# 7 Acknowledgments

We would like to thank Bruno Bombed and Jean-Pascal Dumoulin for their technical help during the
DICASE cruise and in the laboratory. We also thank the captain and crew of the RV Tethys II (INSU)
for their excellent work at sea. A lot of our gratitude is for the SNAP-CO2 for the inter-comparison of
DIC and TA concentrations in our seawater samples. We are grateful to Celine Liorzou for the ICP-
AES measurements, to Serge Miska for the help with the X-Ray diffraction analysis and we want to
thank Stephanie-Duchamp-Alphonse to put a manocalcimeter at our disposition. This research was
financed by the MERMEX project (http://mermex.pytheas.univ-amu.fr/?page_id=62) and by the
MERMEX Rivers project.

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

Van Cappellen, P. and Wang, Y.: Cycling of iron and Manganese in surface sediments: A general theory for the coupled transport and reaction of carbon, oxygen, nitrogen, sulfur, iron and manganese, AM J Sci., 1996.





**Figure captions**

Figure 1: Map of the Rhône River mouth (Northwestern Mediterranean Sea) with the stations investigated during the DICASE cruise in June 2014. The ocean bathymetry is indicated by the continuous lines.

Figure 2: All in situ oxygen micro profiles measured during the DICASE cruise. Profiles measured in
the proximal domain are presented in red, profiles from the prodeltaic domain in blue and profiles from the distal domain in black. The sediment-water interface is marked by a horizontal line (depth = 0).

Figure 3: Diffusive oxygen uptake (DOU) across the sediment-water interface in function of the distance from the Rhône River mouth. The fluxes decrease exponentially following $DOU = F_{min} + Ae^{(-x/t)}$
with $F_{min}$ the flux in the offshore region, x the distance to the river mouth in km. A and t are numerical constants.

Figure 4:   All $pH_t$ micro profiles measured during the DICASE cruise. Profiles measured in the proximal domain are presented in red, profiles from the prodeltaic domain in blue and profiles from the
distal domain in black. The sediment-water interface is marked by a horizontal line.




Figure 5: DIC (black) and TA (red) pore water profiles in the first 30 cm of sediment for the proximal, the prodeltaic and the distal domains.

Figure 6: Calculated $pH_t$ pore water profiles of the proximal, the prodeltaic and the distal domains.

Figure 7: Pore water concentrations of $Ca^{2+}$; proximal domain in red, prodeltaic domain in blue and distal domains in black

Figure 8: Sulfate profiles measured in the pore waters of the proximal, the prodeltaic and in the distal domains and corresponding saturation state of calcium carbonates. The saturation limit for calcium carbonates dissolution/precipitation ($\Omega = 1$) is marked by a vertical line

Figure 9: $\Delta DIC/\Delta SO_4^{2-}$ ratio in the pore waters of the proximal domain corrected for molecular 565 diffusion according to (Berner, 1980).




# Figures

Figure 1

Figure 2



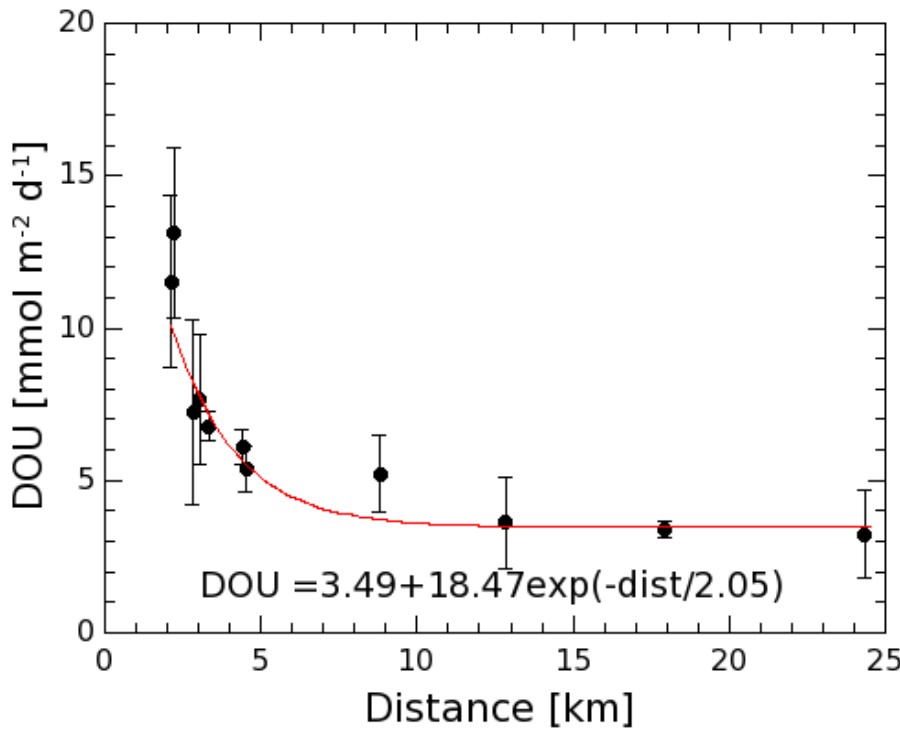

$$DOU = 3.49 + 18.47\exp(-dist/2.05)$$

Figure 3

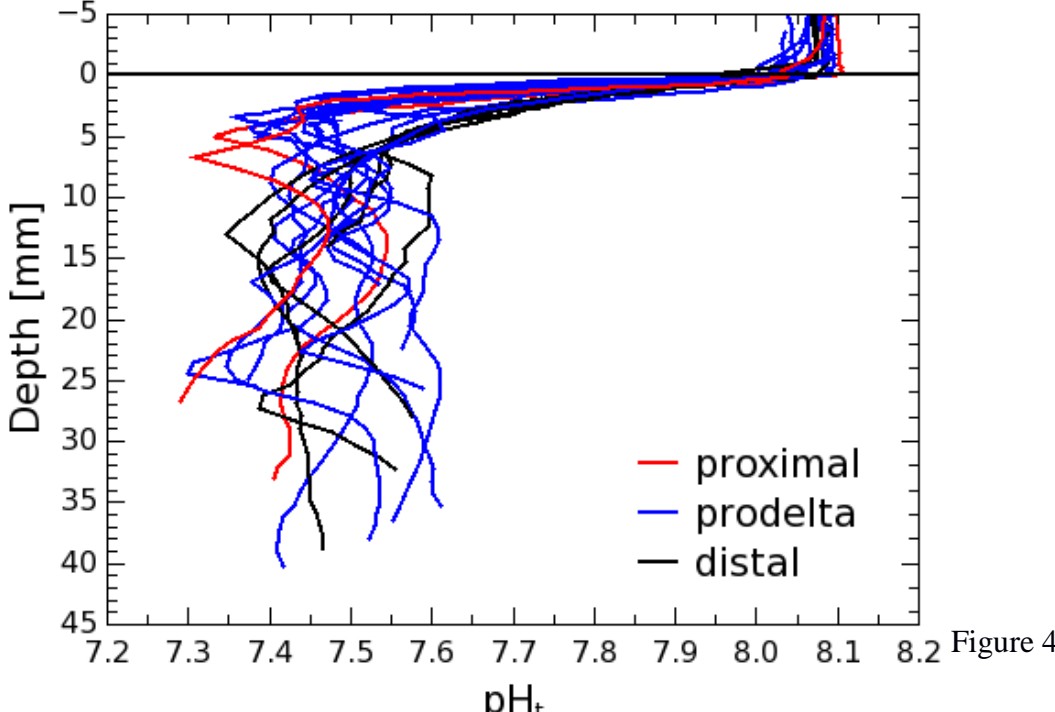

Figure 4



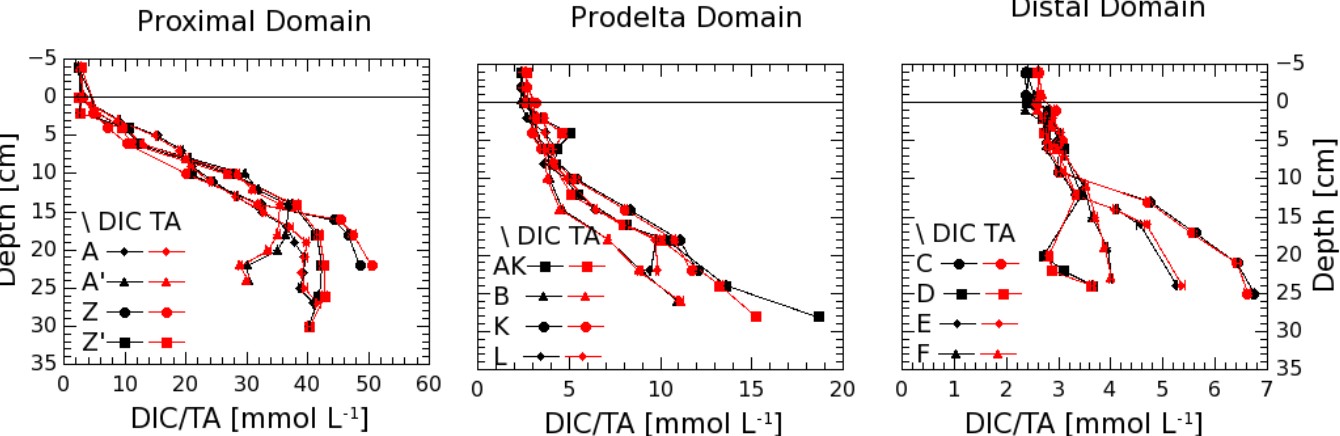

610    Figure 5

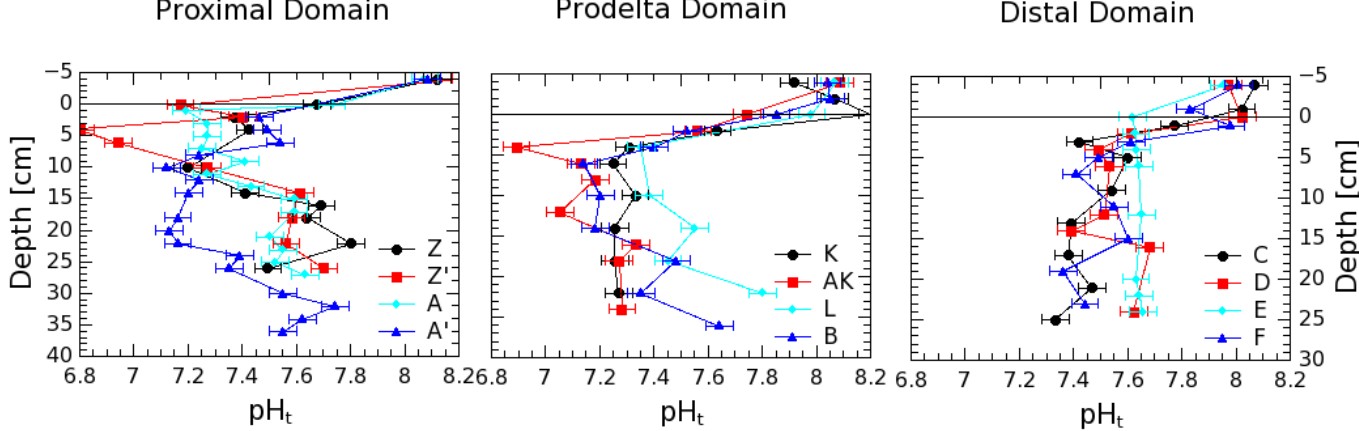

Figure 6

615





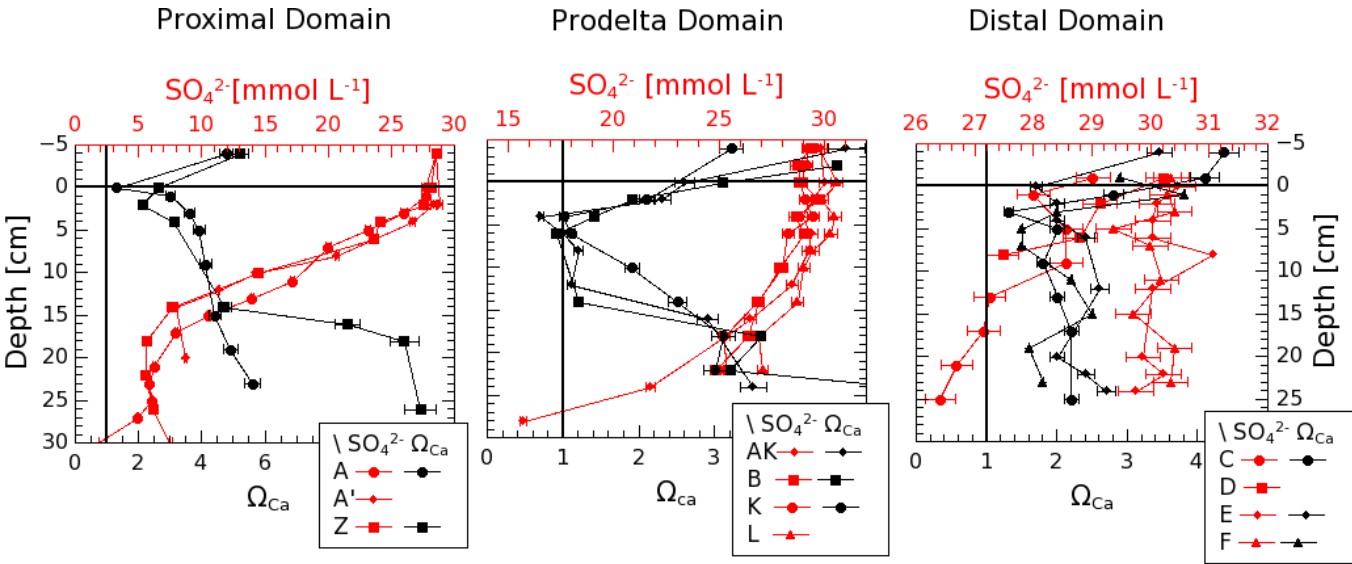

Figure 7

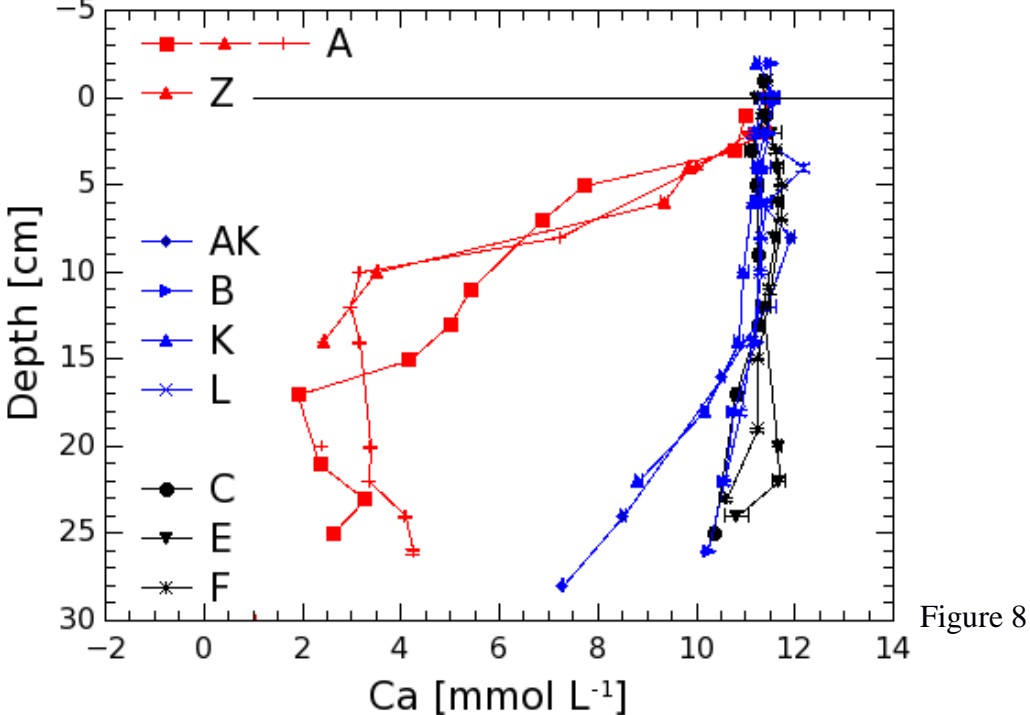

Figure 8





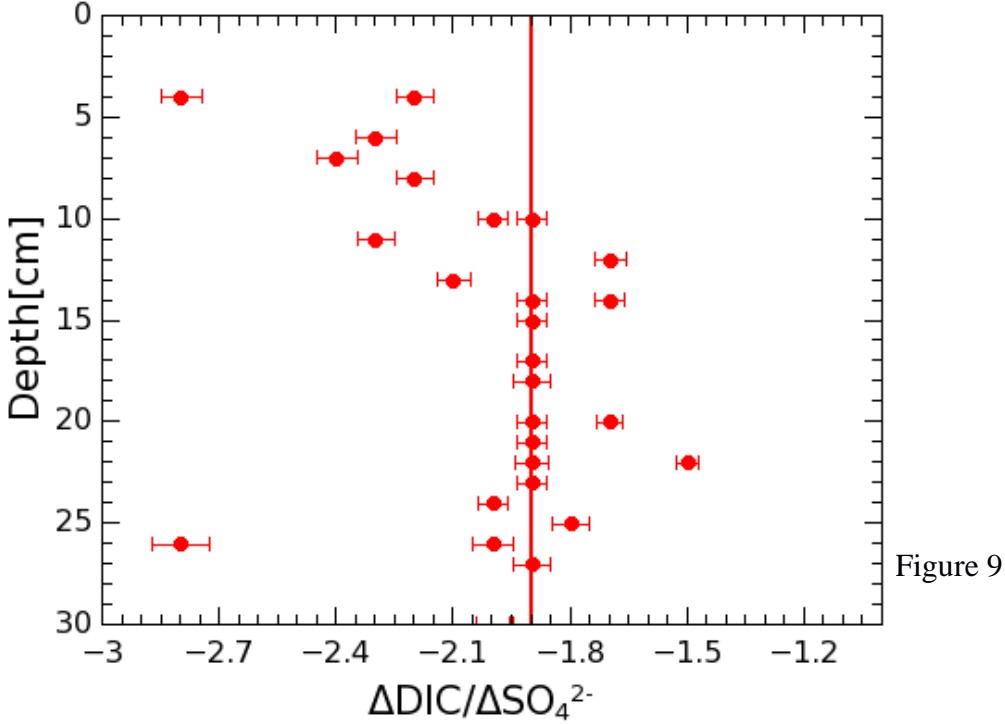

Figure 9





| Station | Long. (°E) | Lat. (°N) | Dist. [km] | Depth [m] | T [°C] | Salinity | O$_2$ [µmol L$^{-1}$] | DIC [µmol L$^{-1}$] | TA [µmol L$^{-1}$] | pH$_t$ | SO$_4^{2-}$ [mmol L$^{-1}$] | pCO$_2$ (calculated) [µatm] |
|---|---|---|---|---|---|---|---|---|---|---|---|---|
| Z, Z' | 4.865 | 43.317 | 2.2 | 18.0 | 16.0 | 37.5 | 244.0 ± 0.3 | 2330 ± 1 | 2648 ± 3 | 8.118 ± 0.003 | 28.4 ± 0.3 | 364.1 |
| A, A' | 4.851 | 43.312 | 2.1 | 18.3 | 16.8 | 37.7 | 245.1 ± 0.3 | 2323 ± 4 | 2613 ± 17 | 8.072 ± 0.004 | 28.2 ± 0.4 | 407.3 |
| AK | 4.856 | 43.307 | 2.8 | 48.1 | 15.8 | 37.4 | 240.8 ± 0.1 | 2335 ± 4 | 2623 ± 3 | 8.085 ± 0.011 | 29.7 ± 0.3 | 393.6 |
| B | 4.818 | 43.295 | 3.0 | 66.2 | 15.0 | 37.7 | 213.2 ± 0.8 | 2372 ± 5 | 2628 ± 2 | 8.039 ± 0.015 | 28.7 ± 0.3 | 446.1 |
| K | 4.856 | 43.302 | 3.3 | 60.5 | 14.9 | 37.7 | 226.4 ± 0.2 | 2351 ± 5 | 2538 ± 5 | 7.916 ± 0.002 | 29.1 ± 0.3 | 596.6 |
| L | 4.885 | 43.304 | 4.4 | 58.2 | 15.2 | 37.6 | 230.9 ± 0.6 | 2340 ± 2 | 2612 ± 5 | 8.066 ± 0.002 | 29.7 ± 0.3 | 412.4 |
| C | 4.773 | 43.271 | 8.8 | 75.0 | 14.4 | 37.7 | 225.6 ± 0.4 | 2354 ± 2 | 2621 ± 10 | 8.067 ± 0.004 | 29.0 ± 0.3 | 411.5 |
| D | 4.738 | 43.256 | 12.8 | 80.0 | 14.9 | 37.6 | 214.5 ± 0.5 | 2388 ± 8 | 2605 ± 3 | 7.970 ± 0.002 | 30.2 ± 0.3 | 531.3 |
| E | 4.685 | 43.219 | 17.9 | 77.3 | 14.3 | 37.7 | 226.3 ± 0.3 | 2391 ± 6 | 2594 ± 5 | 7.952 ± 0.004 | 30.4 ± 0.3 | 553.3 |
| F | 4.649 | 43.164 | 24.3 | 77.0 | 14.8 | 37.7 | 230.2 ± 0.1 | 2364 ± 4 | 2600 ± 24 | 8.008 ± 0.006 | 30.3 ± 0.3 | 478.4 |

640  Table 1: Stations investigated during the DICASE cruise in june 2014 with the main properties of bottom waters; dist = distance from the Rhône river mouth



Table 2: Diagenetic reactions and their effect on the carbonate system

|   | Reaction | $\Delta TA/\Delta DIC$ | $\Delta pH$ | $\Delta\Omega$ |
|---|---|---|---|---|
| R1 | $CO_2 + H_2O \leftrightarrow H_2CO_3 \leftrightarrow HCO_3^- + H^+ \leftrightarrow CO_3^{2-} + 2H^+$ | | - | - |
| R2 | $CaCO_3 + H_2O + CO_2 \leftrightarrow Ca^{2+} + 2HCO_3^-$ | $\pm 2/1$ | + | + |
| R3 | $CH_2O + O_2 \rightarrow HCO_3^- + H^+$ | 0/1 | - | - |
| R4 | $NH_4^+ + 2O_2 \rightarrow NO_3^- + H_2O + 2H^+$ | -2/0 | - | - |
| R5 | $4Fe^{2+} + O_2 + 10H_2O \rightarrow 4Fe(OH)_3 + 8H^+$ | -8/0 | - | - |
| R6 | $2Mn^{2+} + O_2 \rightarrow 2MnO_2$ | | | |
| R7 | $2CH_2O + 4MnO_2 + 6H^+ \rightarrow 2HCO_3^- + 4Mn^{2+} + 4H_2O$ | 8/2 | + | + |
| R8 | $CH_2O + 4Fe(OH)_3 + 7H^+ \rightarrow HCO_3^- + 4Fe^{2+} + 10H_2O$ | 8/1 | + | + |
| R9 | $2CH_2O + SO_4^{2-} \rightarrow 2HCO_3^- + HS^- + H^+$ | 2/2 | - | |
| R10 | $Fe^{2+} + HS^- \rightarrow FeS + H^+$ | -2/0 | - | - |
| R11 | $8Fe(OH)_3 + 9HS^- + 7H^+ \rightarrow 8FeS + SO_4^{2-} + 20H_2O$ | -2/0 | + | + |
| R12 | $8Fe(OH)_3 + 15HS^- + SO_4^{2-} + 17H^+ \rightarrow 8FeS_2 + 28H_2O$ | 2/0 | + | + |
| R13 | $CH_4 + SO_4^{2-} \rightarrow HS^- + HCO_3^- + H_2O$ | 2/1 | | + |
| R14 | $2CH_2O \rightarrow CH_4 + CO_2$ | 0/1 | | |