# Peer review of "Carbonate chemistry in sediment pore waters of the Rhône River delta driven by early diagenesis (NW Mediterranean)"

_Biogeosciences, 2016_

## Author Comment (AC1) · 1 Jun 2016

Dear reader,

In the discussion paper has been an inversion of the figures 7 and 8.

Instead of beeing figure 8, the calcium profiles should be figure 7 (as they are presented first in the text) and the sulfate and saturation profiles should be figure 8 instead of beeing figure 7.

This mistake will be corrected in the revisted version of the paper. I apologize profusely for this confusion,

Jens Rassmann

---

## Referee Comment (RC1) · D. Burdige (Referee) · 15 Jun 2016

In this manuscript Rassmann et al. present an extensive data set on pore water chemistry for sediments from the mouth of the Rhône River to the continental shelf. The data is of high quality and very interesting. Unfortunately, the discussion of the results is too general and poorly focused. Overall I think much more could be done with the data.

Much of the data interpretation is too speculative or is simply based on comparisons with what other workers have seen in these (and other) sediments. In many places the text reads more like a data report interspersed with comments about similarities between these results and results from other studies. The things that are new and exciting and different about this work, as compared to these other studies, are not

clearly presented.

Questions about whether sediments such as these are alkalinity sources is an important one, and the authors note this in places in the text. While they do have some discussions of their results with such considerations in mind, the discussions are rather disjointed. At a bare minimum, Fig. 5 shows that all of these sediments are a source of alkalinity to the water column, although this simple observation seemed (at least to me) to get lost in the overall discussion. I would urge the authors to re-structure the paper so that this general topic is much more clearly examined with their data. In my opinion, this will make this paper one that people will want to read (and should read).

Before final publication the manuscript will need to be carefully copy-edited by a native or fluent English-speaker. There are many places where there are grammatical errors, awkward syntax, and curious phrasings.

One last general comment. When I read lines 91-94 and the sentence starting at the end of line 124 ("The sea floor in this region . . .") I had the sense that these sediments have some degree of similarity to those of other large river deltas like, e.g., the Amazon (see, for example, Aller's 1998 Marine Chemistry paper cited here). In contrast, much of the discussion of the data in the text takes a very traditional, steady-state "Froelich et al."-type approach (see, for example, section 2.7 and much of section 4.1). To me, this approach seems to contradict the text on lines 91-94 and 124, and I think that some clarification is needed.

Specific Comments (line numbers in parentheses)

(215) I never realized there were 12 parameters of the carbonate system. Is this a typo or am I missing something?

(225) Here and on line 291 they talk about good agreement between measured and calculated pH values. It might be good to show this, and/or present some additional information like, e.g., the slope and r2 value of a scatter plot of the two pH's.

(251) I would probably be good to list here what atmospheric pCO2 was at the time of sampling.

(265) The way the pH data is plotted makes it hard to see things like differences in inflection points for different regions. It might be helpful to break Fig. 4 up into 3 panels like Figs. 5 and 6. It might also be useful to similarly sub-divide Fig. 2 (O2 profiles) into 3 panels.

(286) Are these slopes statistically different in the three different regions? If not I would not report them separately but would simply list an overall slope for all of the sediments.

(317 -) Plotting sulfate concentrations and carbonate saturation state for each region on the same panels is very confusing. I would recommend separating them.

(405-) I would think that all of the things discussed here (organic matter oxidation state, carbonate precipitation, AOM) would affect the magnitude of the slope of a $\Delta$DIC/$\Delta$Sulfate plot, and not the scatter around the best-fit line. I'm also surprised that the slope is $\sim$2 despite all of these factors. Maybe they act (somehow) in such a way as to cancel each other out?

(473) I don't see any direct evidence in the paper that terrestrial organic matter is what is being degraded. It might be, but without evidence to support this I would not be so definitive.

---

## Referee Comment (RC2) · Anonymous Referee #2 · 24 Jul 2016

Rassmann and collaborators present a very nice dataset of sediment properties in the Rhône river delta. Based on direct (microelectrodes) measurements of pH and O2 and on pore-water analyses of DIC, TA, Ca2+ and SO42- along a gradient from the river mouth to the open Mediterranean Sea (3 domains considered), this manuscript aims to describe and understand the main diagenetic reactions that control these sediment properties and the impact of the sediment on the bottom water carbonate chemistry. I would recommend publication of this manuscript following the proposed minor modifications and an extensive copy-edition by a native speaker.

Table 2 and Section 2.7 should not be presented in the Material and Methods section but more likely in the Discussion. I do not believe CO2 dissolution should be presented

as a diagenetic reaction. Table 2 should be made much clearer and for instance updated by: 1) providing the full name of the presented reactions, 2) dividing into 3 parts with reactions occurring in the presence of oxygen (oxic mineralization and reoxidation of reduced species), in the anoxic section (anaerobic mineralization and precipitation of reduced species), and both: CaCO3 dissolution or precipitation. Reaction R2, it is not clear whether you present CaCO3 dissolution or precipitation or both. I would consider 2 lines, one for precipitation, and one for dissolution as their effects on TA, DIC, pH and Omega are opposite. Finally, I would move this reaction to the end of the table (see above). Why don't you show nitrate reduction? This can be in some cases an important pathway. All reactions should be presented considering the same amount of OM mineralized (in some cases you have 1 or 2 moles of CH2O mineralized).

Figure 6 should be updated. First, you don't show the same Y-axis scale than on Figure 5, why is that? and furthermore, this scale in not the same between the 3 domains in this Figure 7. I am a bit surprised by the very high heterogeneity that you found between stations in the Proximal domain and believe there are a number of mistakes to correct. For instance, you have DIC and TA data for station Z' until 30 cm while you calculate pH up to 25. For station A', you seem to have DIC/TA data down to 25 cm and you calculate pH data down to 35 cm or more. Please check. Furthermore, I have calculated pH for station Z at the last sampled depth (between 20 and 25 cm) considering TA of 48 and DIC of 50 mmol/L, I end up with a pH of 7.18, far from the 7.8 shown in Figure 6. Again, this should be carefully checked. L284-287: You should present the determination coefficients and the corresponding slopes for each domain separately. L291: I would really like to see a more detailed comparison between measured and calculated pH. Section 3.4. Why all stations are not shown for Ca2+ (D, A' and Z' missing), why 3 datasets for A ?

Minor corrections: L39: et al., L43: This is not correct, following your Table 2, aerobic mineralization does not produce TA. L54: Jahnke et al. (1997) L57: (Jahnke and Jahnke, 2004) L62: Jahnke et al. (1997) and Jahnke and Jahnke (2004). Burdige

et al. (2008, 2010) L65: Van Capellen and Wang (1996) L69: (Froelich et al., 1979; Berner, 1980) L72: Pastor et al., 2011 a or b? L72: (Jourabchi et al. 2005; . . .) L80: (Arndt et al., 2013; . . .) L86: ; Cathalot et al., 2013) L92: Cathalot et al., 2010 L110: Abbrevation for year should be consistent throughout the text (yr-1) L114: Roussiez et al., 2005 L119: Got and Aloisi (1990) L124: Miralles et al., 2005 L141: in situ in talics L152: I don' understand what is "the slope of the pH variation", please rephrase. L156: .. "using this BW value the micro electrode measured pH variations", please rephrase L162: surface of 109 cm, this is not a surface. L171: according to Broecker and Peng (1974) L180 to 191: please precise how many replicates were measured for each parameter (pH, O2, DIC and TA) L254: oxygen penetration depth (OPD) was of: please correct L259-260: units should be mmol m-2 d-1 L265: between 7.4 AND 7.6 L267: please add high "spatial" heterogeneity L299: (Fig. 5) L313: > 95% calcite + >5 % Mg-calcite do not leave much room for aragonite. . ..

Many grammatical and formatting errors in the Discussion.

All figures starting from Figure 2: Average OPD for each domain should be shown on these plots. Figure 2 legend: in situ in italics Figure 3 legend: what do the vertical error bars correspond to? Please add.

Figures 5 and 7: I would use the same x-axis scale for all 3 domains and for Figure 7 separate SO42- and Omega, consider using 2 figures. On Figure 7, please note that the legend box hides the axis, which should be avoided.

Figure 6 and 7, and legends: what do the horizontal error bars correspond to? Please add.

---

## Author Comment (AC3) · 5 Sep 2016

We thank the reviewer for his helpful and constructive comments. In the following document, we answer the questions one by one. Modifications that have been done in to the manuscript are written in *italics*.

All minor comments of Reviewer 2 were taken into account.

**Referee:** Rassmann and collaborators present a very nice dataset of sediment properties in the Rhône river delta. Based on direct (microelectrodes) measurements of pH and O2 and on pore-water analyses of DIC, TA, Ca2+ and SO42- along a gradient from the river mouth to the open Mediterranean Sea (3 domains considered), this

manuscript aims to describe and understand the main diagenetic reactions that control these sediment properties and the impact of the sediment on the bottom water carbonate chemistry. I would recommend publication of this manuscript following the proposed minor modifications and an extensive copy-edition by a native speaker.

**Answer:** The manuscript was copy-edited by a native English-speaker (Patrick Laceby, LSCE)

**Referee:** Table 2 and Section 2.7 should not be presented in the Material and Methods section but more likely in the Discussion.

**Answer:** We moved Section 2.7 to the beginning of the discussion and rephrased it.

**Referee:** I do not believe CO2 dissolution should be presented as a diagenetic reaction. Table 2 should be made much clearer and for instance updated by: 1) providing the full name of the presented reactions, 2) dividing into 3 parts with reactions occurring in the presence of oxygen (oxic mineralization and reoxidation of reduced species), in the anoxic section (anaerobic mineralization and precipitation of reduced species), and both: CaCO3 dissolution or precipitation. Reaction R2, it is not clear whether you present CaCO3 dissolution or precipitation or both. I would consider 2 lines, one for precipitation, and one for dissolution as their effects on TA, DIC, pH and Omega are opposite. Finally, I would move this reaction to the end of the table (see above).

**Answer:** We sub-divided Table 2 into the 3 suggested categories and split reaction 2 into two lines for dissolution and precipitation.

**Referee:** Why don't you show nitrate reduction? This can be in some cases an important pathway.

**Answer:** Waters of the Northwest Mediterranean Sea show low nitrate concentrations. Only a minor part of OM is mineralized by this pathway in the study area. Nevertheless, denitrification has been added to the reaction table. To justify not to discuss nitrate reduction in this area, we cite (Pastor, L., Cathalot, C., Deflandre, B., Viollier,

E., Soetaert, K., Meysmann, F.J.R., Ulses, C., Metzger, E. and Rabouille, C.: Modeling biogeochemical processes in sediments from the Rhône River prodelta area (NW Mediterranean Sea), Biogeosciences, 8, 1351-1366, 2011a.) at line 387:

*"In contrast to other nearshore environments, nitrate reduction has been shown to account only for 2-5 % of OM mineralization in the sediments of the prodelta of the Rhône whereas other anaerobic mineralization processes account for 30-40 % in the distal domain and up to 90 % in the proximal domain (Pastor et al., 2011a). Nitrate reduction produces less TA than DIC (TA/DIC ratio = 0.8/1) and thus lowers $\Omega$."*

**Referee:** All reactions should be presented considering the same amount of OM mineralized (in some cases you have 1 or 2 moles of CH2O mineralized).

**Answer:** The stoichiometric coefficients in table 2 have been adjusted to 1 mol of $CH_2O$.

**Referee:** Figure 6 should be updated. First, you don't show the same Y-axis scale than on Figure 5, why is that? and furthermore, this scale in not the same between the 3 domains in this Figure 7.

**Answer:** We adjusted the scale on all figures to 40 cm depth.

**Referee:** I am a bit surprised by the very high heterogeneity that you found between stations in the Proximal domain and believe there are a number of mistakes to correct. For instance, you have DIC and TA data for station Z' until 30cm while you calculate pH up to 25. For station A', you seem to have DIC/TA data down to 25 cm and you calculate pH data down to 35 cm or more. Please check.

**Answer:** In effect, the area is highly heterogeneous, differences of 10 mmol/L in DIC or TA pore water concentrations at a same station in a certain depth are definitely possible. We checked the figures for consistency. Indeed, some data points had disappeared on the graphs and were re-introduced.

**Referee:** Furthermore, I have calculated pH for station Z at the last sampled depth (between 20 and 25 cm) considering TA of 48 and DIC of 50 mmol/L, I end up with a

pH of 7.18, far from the 7.8 shown in Figure 6. Again, this should be carefully checked.

**Answer:** We use CO2SYS with the thermodynamic constants from (Lueker, T. J., Dickson, A. G., Keeling, C. D.: Ocean $pCO_2$ calculated from dissolved inorganic carbon, alkalinity, and equations for K1 and K2 : validation based on laboratory measurements of $CO_2$ in gas and seawater at equilibrium, Mar. Chem., 70, 105-119, 2000.) with TP = 0.1 $\mu$mol kg$^{-1}$ and TSi = 6.4 $\mu$mol kg$^{-1}$ (Denis, L., Grenz, C.: Spatial variability in oxygen and nutrient fluxes at the sediment-water interface on the continental shelf in the Gulf of Lions (NW Mediterranean), Oceanologica Acta, 26, 373-389, 2003) and making the hypothesis that theses values are constant with depth (in lack of better data). Furthermore, we use the bottom water salinity of 37.5 and the in situ temperature of 16.0 °C. The water depth at this station is 18 m. The analysis has been done at a temperature of 25 °C.

For the data point cited:
Z(18 cm): TA = 46.215 $\pm$ 0.474 $\mu$mol kg$^{-1}$, DIC = 45.440 $\pm$ 0.190 $\mu$mol kg$^{-1}$ leads to pH= 7.636 $\pm$ 0.059
Z(22 cm): TA = 49.189 $\pm$ 0.504 $\mu$mol kg$^{-1}$, DIC = 47.363 $\pm$ 0.129 $\mu$mol kg$^{-1}$ leads to pH= 7.801 $\pm$ 0.048 and
Z(26 cm): TA = 44.514 $\pm$ 0.484 $\mu$mol kg$^{-1}$, DIC= 44.506 $\pm$ 0.051 $\mu$mol kg$^{-1}$ leads to pH=7.492 $\pm$ 0.051

If the discrepancies of your and our calculations persist, we should do a more detailed comparison of our methods with a whole dataset.

**Referee:** L284-287: You should present the determination coefficients and the corresponding slopes for each domain separately.

**Answer:** As the slopes were not significantly different, we followed the suggestion of Reviewer 1 and reported the overall slope 1.01 with $r^2$ = 0.9982.

**Referee:** L291: I would really like to see a more detailed comparison between measured and calculated pH.

**Answer:** As already posted in the answer to Reviewer 1:

We added at line 292 : *"A linear relationship of the pH data measured with microelectrodes against calculated pH by CO2SYS shows a correlation with a slope of 1.01 ± 0.02 and an $r^2$ = 0.7483 (graph not shown)."*

We compared the pH values calculated from DIC and TA data with microelectrode data. As the porewaters represent an integration of a certain sediment zone, the average signal of the microelectrodes for the same zone was used. The size of the influenced zone was calculated following: Seeberg-Elverfeldt, J., Schlüter, M., Feseker, T., Kölling, M.: Rhizon sampling of porewaters near the sediment-water interface of aquatic systems; Limnology and Oceanography: Methods, 3, 361-371, 2005

Because we already include 10 figures, we did not include a figure showing the correlation. In this answer, the correlation is visible on Figure 1.

**Referee:** Section 3.4. Why all stations are not shown for Ca2+ (D, A' and Z' missing), why 3 datasets for A ?

**Answer:** We do not have any data for stations D and Z'. In fact, there has been a confusion at station A: the data sets are from A and A' and a longer core at station A that has been removed in the reviewed version of the paper.

**Referee:** L43: This is not correct, following your Table 2, aerobic mineralization does not produce TA.

**Answer:** We rephrased:

*"Aerobic and anaerobic reaction pathways contribute to the production of dissolved inorganic carbon (DIC), which creates acidification of the bottom waters. Anaerobic reactions lead as well to production of total alkalinity (TA)"*

**Referee:** L152: I don' understand what is "the slope of the pH variation", please

rephrase.

**Answer:**We rephrased:

*"The calibration of the pH electrodes was carried out using NBS buffers, thus allowing the estimation of the slope of the electrode signal in fonction of pH variation at onboard temperature. The slope was then recalculated at in situ temperature and the electrode signal variation was transformed into pH changes. "*

**Referee:** L156: .. "using this BW value the micro electrode measured pH variations", please rephrase

**Answer:**We rephrased:

*"The pH of bottom waters was determined using the spectrophotometric method with m-cresol purple following Clayton and Byrne, (1993) and Dickson et al., (2007). Pore water pH on the total proton scale (pHt) was recalculated using the signal of the micro-electrode adjusted to this BW value."*

**Referee:** L162: surface of 109 cm, this is not a surface.

**Answer:** The surface of the head of the moving unit of the lander, where the electrodes are mounted measures 109 $cm^2$.

**Referee:**L171: according to Broecker and Peng (1974) L180 to 191: please precise how many replicates were measured for each parameter (pH, O2, DIC and TA)

**Answer:** *"All bottom water concentrations were measured as triplicates. Small sample volumes in pore waters only allowed for replicates for the DIC, $SO_4^{2-}$ and $Ca^{2+}$ analysis."*

**Referee:** L299: (Fig. 5) L313: $> 95\%$ calcite + $>5$ % Mg-calcite do not leave much room for aragonite: : :.

**Answer:**Yes, indeed, we did not detect any aragonite in this area. The drainage basin

of the Rhône River is characterized by old carbonates and the calcifying organisms on site (foraminifers) produce calcite tests. There are no corals in this area.

In lack of better material, the aragonite standard used for the calibration of the X-ray analysis is natural coral powder, characterized to have less than 2 % of Ca content. (Kindler, P., Reyss, J.-L., Cazala, C., Plagnes V.: Discovery of a composite reefal terrace of middle and late Pleisocene age in Great Inagua Island, Bahamas. Implications for regional tectonics and sea-level history, Sedimentary Geology, 194, 141-147, 2007). It's diffraction analysis gives the following diffractogramme (Figure 2) with the clearly visible principal Ar peak at $2\theta = 26.297°$ and its secondary peak at $27.298°$ (both slightly shifted from the theoretical value) .

A typical diffractogramme of Rhone delta sediments looks like Figure 3 (Station Z). We recognize clearly the principal calcite peak at $2\theta = 29.468°$ with a little deformation at the right indicating the presence of magnesian calcite. The position of the aragonite peak is covered by the base of another peak at $2\theta = 26.674°$. These diffractogrammes do not allow a better deconvolution of the peaks and we have an uncertainty about the material used to calibrate our DRX measurements. So the presence of aragonite cannot be completely excluded, but is unlikely.

**Referee:** Many grammatical and formatting errors in the Discussion.

**Answer:** Grammatical and formatting errors have been corrected.

**Referee:** All figures starting from Figure 2: Average OPD for each domain should be shown on these plots.

**Answer:**The vertical scale on figures 2 and 4 is in mm whereas the vertical scale on the following figures is given in cm. The average OPD for each domain is situated between the sediment water interface and the first scale trait, so it would be invisible on these figures.

**Referee:**Figure 2 legend: in situ in italics Figure 3 legend: what do the vertical error

bars correspond to? Please add.

**Answer:**We added in the legend of the figure:

*"Error bars are standard deviations between the diffusive fluxes calculated from the 5 single oxygen profiles measured at each station."*

**Referee:**Figures 5 and 7: I would use the same x-axis scale for all 3 domains

**Answer:**We decided not to use the same concentration scale because the gradients are very different. Adjusting the scale for all figures would hide the form of the profiles in the prodelta and distal domain. To alert the reader about difference between the three concentration scales, we added to the figure caption:

*"For better visibility of the profiles in each domain, the scale of the concentrations has been individually adjusted for each domain."*

**Referee:**and for Figure 7 separate SO42- and Omega, consider using 2 figures. On Figure 7, please note that the legend box hides the axis, which should be avoided.

**Answer:**The figure has been split into individual figures for sulfate concentrations and calcium carbonate saturation states.

**Referee:** Figure 6 and 7, and legends: what do the horizontal error bars correspond to? Please add.

**Answer:**We added to the caption of figure 6:

*"The error bars are estimations of the error propagation from the measured data in the calculation."*

**Answer:**and to the caption of figure 7:

*"The error bars represent the uncertainties of the ICP-AES measurement."*

[Figure]

slope: 1.01 ± 0.02
$r^2$ : 0.7483

[Scatter plot with x-axis labeled "pH-microelectrodes" ranging from 6,8 to 8,4 and y-axis labeled "pH-calculated" ranging from 6,8 to 8,4]

**Fig. 1.**

**Fig. 2.**

**Fig. 3.**

---

## Author Response (AR2)

**Manuscript revision (doi:10.5194/bg-2016-212): Responses to the Associate Editors comments:**

We thank Jack Middelburg for his helpful and constructive comments that we have taken into account for our manuscript.

L. 14 and all through: use "pore-water" with hyphen if an adjective

We added a hyphen.

L. 15: reformulate TA is not a concentration

Indeed, TA is the equivalent of the weak bases in seawater.
We reformulated:
TA and concentrations of DIC, $SO_4^{2-}$ and $Ca^{2+}$ were analyzed on bottom waters and extracted sediment pore waters, whereas pH and oxygen concentrations were measured *in situ* using microelectrodes.

L. 19: delete aerobic: you cannot see the difference between re-oxidation and aerobic respiration from OPD.

We deleted the word aerobic.

L. 22: aerobic processes, oxic conditions: hence use aerobic here

The word oxic was replaced by the word aerobic.

L. 74: I do not understand this. The knowledge presented in Berners' 1980 book is sufficient to explain all this. Reformulate.

We reformulated the sentence:
 These authors, with Froelich et al. (1979) and Berner (1980) as precursors,  highlighted the complexity of the multiple competing reaction pathways in anoxic sediment.

L. 101: delete differs

We deleted the word differs.

L. 140: sediment accumulation or sediment deposition?

In contrast to the systems like the Amazonas delta, the sediment in the prodelta of the Rhône is not caracterized by intense reworking and fluidized beds, but characterized by high deposition rates that lead to high accumulation rates. We reformulated:
The seafloor in this region is a dynamic environment based on sediment deposition and accumulation and with important heterogeneity regarding diagenetic activities, sediment pore water profiles and exchange fluxes at the sediment-water interface (Lansard et al., 2009; Cathalot et al., 2010).

L. 283/284: TA is not a concentration: rewrite

We reformulated into : DIC concentrations and TA

L. 299: Existing numerical tools have been developed…. Water column. Nevertheless we used…….
Is the neglect of metabolites in TA calculation not the real problem?

We rephrased:
Existing numerical tools have been developed for the oxic water column. Nevertheless we used them in the sediments knowing that pore water concentrations (e.g. DIC, TA) are much larger than those in the water column and that reduced metabolites are neglected.

L. 313: indicate high sulfate..

We added a subordinate:
The distal domain is characterized by constant $Ca^{2+}$ concentrations which remain above 10 mmol $L^{-1}$and the sulfate concentrations stay close to bottom water concentration.

L. 334-352: this whole section needs referencing. These reaction have been worked out in detail by Boudreau, Soetaert and others.

References were added to the whole paragraph.

L. 360: denitrification pathway does not involved re-oxidation because N2 escapes and thus contributes disproportionally.

We added:
Denitrification forms an exception to this, as the reduced end products $N_2$ of this reaction can escape the sediments. Thus the TA produced by denitrification is not consumed in the oxic sediment layer.

L. 372: I suggest deleting aerobic. Not needed and geochemists have since appreciated more re-oxidation.

We deleted the word aerobic.

L. 435: Denitrification section should mention that N2 is not oxidized and thus TA can escape.

We added:
Nevertheless denitrification is a net TA source from the sediments to the bottom water as the end product $N_2$ undergoes no further oxidation.

References list: my name is twice misspelled: Arndt et al. reference and Soetaert et al. reference.
We are sorry for the mistake and corrected the spelling. Other references were added.

Table 2: please check your reactions: R6 is unbalanced and its effect on TA, saturation degree, DIC etc is not mentioned

The equation has been balanced and the effects on TA, DIC and Ω have been added:

| R6 | Manganese oxidation | $2Mn^{2+} + O_2 + 4HCO_3^- \rightarrow 2MnO_2 + 4CO_2 + 2H_2O$ | -4/0 | - | - |

Also R15 has an effect on pH but that is complex function of pH (see Soetaert et al. 2007).

In effect, R15 drives pH towards the value of 7.9. So we precised:

| R15 | Anaerobic methane oxidation | $CH_4 + SO_4^{2-} \rightarrow HS^- + HCO_3^- + H_2O$ | 2/1 | + (if pH < 7.9) | + |

**Manuscript revision (doi:10.5194/bg-2016-212): Responses to the referees' comments:**

We thank the reviewer for his helpful and constructive comments. In the following document, we answer the questions (in black) one by one in blue. Modifications that have been done in to the manuscript are written in *italics*.

**Reviewer #1** D. Burdige (Referee)

Unfortunately, the discussion of the results is too general and poorly focused. Overall I think much more could be done with the data. Much of the data interpretation is too speculative or is simply based on comparisons with what other workers have seen in these (and other) sediments. In many places the text reads more like a data report interspersed with comments about similarities between these results and results from other studies. The things that are new and exciting and different about this work, as compared to these other studies, are not clearly presented.

In the reviewed version, we focus more on the new findings of our work. For this area, we present the first data set containing at once pore water concentrations of DIC, TA and microprofiles of oxygen and pH in order to deal with carbonate dissolution/precipitation in these delta sediments. But as diagenetic processes are very complex, we were not able to measure all parameters. For this reason, sometimes we have no other choice than to rely on literature data and hypothesis.
Concerning the importance of the presented work, the following sentences have been added to the abstract (line 29):

*The large production of pore water alkalinity characterizes these sediments as an alkalinity source to the water column which may increase the $CO_2$ buffering capacity of these coastal waters. Estuarine sediments should receive more attention in future estimations of global carbon fluxes.*

To differentiate our work from previous studies, we added a couple of lines in the introduction (line 94):

*Previous studies in this region often focused on organic matter mineralization pathways measurements in the oxic sediment layers and analysis of particulate carbon (Lansard et al., 2008; Cathalot et al., 2010) or could not provide simultaneous DIC and TA pore water measurements (Pastor et al., 2011a). These studies did not provide information on TA production and fluxes at the SWI. Accordingly, we designed a study to investigate the interaction of mineralization processes on porewater pH and the fate of solid calcium carbonates. For that purpose, we used a combination of in situ oxygen and pH microelectrode measurements and pore water analysis of DIC, TA, $SO_4^{2-}$ and $Ca^{2+}$ concentrations to examine various diagenetic pathways on different vertical scales.*

Questions about whether sediments such as these are alkalinity sources is an important one, and the authors note this in places in the text. While they do have some discussions of their results with such considerations in mind, the discussions are rather disjointed. At a bare minimum, Fig. 5 shows that all of these sediments are a source of alkalinity to the water column, although this simple observation seemed (at least to me) to get lost in the overall discussion. I would urge the authors to re-structure the paper so that this general topic is much more clearly examined with their data. In my opinion, this will make this paper one that people will want to read (and should read).

Indeed this question is very important and we modified the text to focus the article more in this direction. To better introduce this question, we added in the introduction (line 44) ... :

*Anaerobic reactions also lead to production of total alkalinity (TA) that increases the $CO_2$ buffer capacity of seawater (Thomas et al., 2009). Variations in DIC and TA affect the partial pressure of $CO_2$ ($pCO_2$) in seawater and ultimately the $CO_2$ exchange with the atmosphere (Emerson and Hedges, 2008). By increasing the $CO_2$ buffer capacity of seawater, the release of TA from anaerobic sediments into the water column could account for a majority of the $CO_2$ uptake in shelf regions and deliver as much TA to the oceans as is derived from rivers (Thomas et al., 2009). Due to high dynamics, spatial heterogeneity and complex biogeochemical mechanisms, estimations of TA fluxes from the sediments are affected by high uncertainties (Krummins et al., 2013).*

... and discussed more explicitly in new paragraph between lines 325 and 326:

*In the Rhône River delta sediments, OM mineralization leads to DIC production, and under anoxic conditions, also to TA production. Our results demonstrate strong DIC and TA pore water gradients in the anoxic layer of the sediments indicating high anaerobic respiration rates. As a result, DIC and TA diffuse towards the SWI. No oxic reaction consumes DIC except potential carbonate precipitation. Our results indicate that more DIC is produced in the sediments than consumed by precipitation of $CaCO_3$. This means, that OM mineralization in the sediments leads to strong DIC fluxes from the sediments into the water column. For TA, the situation is more complicated, as oxidation of reduced species can consume as much TA as has been produced to reduce these species (Table 2). In a 1D system, where no precipitation occurs and no reduced species can be exported, 100% of the anaerobic TA would be consumed in the oxic layer.*
*Krumins et al., (2013) reported that the effective TA flux from the sediments into the water column is far less important than the anaerobic TA production due to the TA loss in the oxic layer. Unfortunately, the resolution of the DIC and TA pore water profiles in this current study does not give precise information about the gradients in the oxic layer. Thus, we can only speculate about the oxic TA consumption in this region and related TA fluxes across the SWI. According to (Pastor et al., 2011a), 97 % of the reduced species precipitate in the anoxic sediments in the Rhône prodelta. Therefore, the majority of the produced TA is likely released into the water column which can counterbalance the effects of the DIC fluxes and increase the $CO_2$ buffer capacity of the overlaying waters.*

and rephrased from line 455 to the end :

*As the alkalinity fluxes produced by anaerobic processes are high and likely not much reduced by reoxidation of reduced species in the oxic layer due to iron sulfide precipitation, net TA fluxes of the same order of magnitude than DIC fluxes are likely to occur. Therefore, the alkalinity build up in the anoxic zone could diffuse across the oxic sediment layer and contribute to buffer bottom waters and increase $CO_2$ storage capacity of these waters. The large precipitation of calcium carbonate in the proximal zone may have implications for the $CO_2$ source potential from the sediment. Indeed, calcium carbonate precipitation generates $CO_2$ (R2b) which can then be exported to the water column. In addition, calcium carbonate precipitation consumes TA. However, the order of magnitude of the TA consumption by carbonate precipitation in these sediments is below the quantity of TA produced by sulfate reduction. Without this TA flux, the $pCO_2$ of the bottom waters in the prodelta of the Rhône would likely be much higher than observed.*

In order to make sure if these sediments are important alkalinity sources or if the majority of the anaerobically produced TA is consumed in the oxic sediment layer, we carried out in situ flux

measurements together with Martial Taillefert and Eryn Eitel in September 2015. The data wich are still beeing processed will be published in a separate paper.

Before final publication the manuscript will need to be carefully copy-edited by a native or fluent English-speaker. There are many places where there are grammatical errors, awkward syntax, and curious phrasings.

The manuscript was copy-edited by a native English-speaker (Patrick Laceby, LSCE)

One last general comment. When I read lines 91-94 and the sentence starting at the end of line 124 ("The sea floor in this region : : :") I had the sense that these sediments have some degree of similarity to those of other large river deltas like, e.g., the Amazon (see, for example, Aller's 1998 Marine Chemistry paper cited here). In contrast, much of the discussion of the data in the text takes a very traditional, steady-state "Froelich et al."-type approach (see, for example, section 2.7 and much of section 4.1). To me, this approach seems to contradict the text on lines 91-94 and 124, and I think that some clarification is needed.

Indeed, this environment is very dynamic, but very different from the Amazon delta due to the lack of tidal mixing and strong permanent currents. The prodelta of the Rhône is dominated by very high accumulation rates due to flood depositions and resuspension events during winter stroms can remove several centimeters of sediment. Despite this fact, molecular diffusion is the dominant transport process and we find the same general tendencies from cruise to cruise.

We added a sentence to the description of the sampling site to point at the particularity of this environment at line 92 and rephrased:

*The "predominance" of sediment accumulation over other dynamic processes and the absence of tidal mixing and dominant marine currents differentiate the prodelta of the Rhône differs from other deltaic environments like the Amazon, where the surface sediments are constantly reworked (Aller et al., 1998).*

To discuss different transport mechanisms, we added at the end of the section 4.1 (line 387) :

*Finding this clear succession of reactions is interesting, particularly the pH profiles that look classical in the aerobic sediment layers sampled from this complex and dynamics system. As OPDs measure only a couple of mm, molecular diffusion is by far the dominant transport process (Peclet number $\gg 1$ on a scale of the OPD). The microstructure of these sediments is restored very fast after distrubances like resuspension events (Toussaint et al., 2014). Furthermore, the comparison with previous studies shows, that despite the high sediment dynamics in this region, the general biogeochemical tendencies are maintained throughout time.*

and added a new reference:
*Toussaint F., Rabouille, C., Cathalot, C., Bombled, B., Abchiche, A., Aouji, O., Buchholtz, G., Clemençon, A., Geyskens, N., Répécaud, M., Pairaud, I., Verney, R. and Tisnérat-Laborde, N.: A new device to follow temporal variations of oxygen demand in deltaic sediments: the LSCE benthic station, Limnol. Oceanogr.: Methods, 12, 729-741, 2014*

Specific Comments (line numbers in parentheses)

(215) I never realized there were 12 parameters of the carbonate system. Is this a typo or am I missing something?

Sorry, there are 9  parameters, the mistake has been corrected.

(225) Here and on line 291 they talk about good agreement between measured and calculated pH values. It might be good to show this, and/or present some additional information like, e.g., the slope and r2 value of a scatter plot of the two pH's.

We added at line 292 :
*A linear relationship of the pH data measured with microelectrodes against calculated pH by CO2SYS shows a correlation with a slope of 1.01 +/- 0.02 and an r² = 0.7483 (graph not shown).*

We compared the pH values calculated from DIC and TA data with microelectrode data. As the porewaters represent an integration of a certain sediment zone, the average signal of the microelectrodes for the same zone was used. The size of the influenced zone was calculated following: Seeberg-Elverfeldt, J., Schlüter, M., Feseker, T., Kölling, M.: Rhizon sampling of porewaters near the sediment-water interface of aquatic systems; Limnology and Oceanography: Methods, 3, 361-371, 2005

[Figure]

Because we already include 10 figures, we did not include a figure showing the correlation.

(251) I would probably be good to list here what atmospheric pCO2 was at the time of sampling.

We added at line 251:

*During the sampling period, the Integrated Carbon Observation System (ICOS) station at Manosque (l'Observatoire de Haute Provence, https://icos-atc.lsce.ipsl.fr/?q=OHP) measured a pCO2 of 410 ppm. At most stations, pCO$_2$ was oversaturated compared to the atmosphere, with the lowest values calculated close to the river mouth at stations A and Z and the highest values calculated in the bottom waters at the shelf stations*

(265) The way the pH data is plotted makes it hard to see things like differences in inflection points for different regions. It might be helpful to break Fig. 4 up into 3 panels like Figs. 5 and 6. It might also be useful to similarly sub-divide Fig. 2 (O2 profiles) into 3 panels.

Figure 2 and Figure 4 were sub-divided into 3 panels.

(286) Are these slopes statistically different in the three different regions? If not I would not report them separately but would simply list an overall slope for all of the sediments.

We changed the corresponding sentence at line 286 into:

*The DIC and TA pore water profiles are well correlated in each core and the concentrations show a linear correlation with a slope of 1.01 and an r² = 0.9982 (130 data points).*

(317 -) Plotting sulfate concentrations and carbonate saturation state for each region on the same panels is very confusing. I would recommend separating them.

The figure in question was subdivided into two figures showing sulfate profiles and saturation states separately.

(405-) I would think that all of the things discussed here (organic matter oxidation state, carbonate precipitation, AOM) would affect the magnitude of the slope of a DIC/Sulfate plot, and not the scatter around the best-fit line. I'm also surprised that the slope is 2 despite all of these factors. Maybe they act (somehow) in such a way as to cancel each other out?

We changed the representation of the results into a $\Delta DIC$ vs $\Delta SO_4^{2-}$ plot that has a slope of 1.65. Taking into account the difference of the diffusion speed in sediments of these two species, we come close to a ratio of 2. Indeed, this is very surprising and we think that the processes in question cancel each other out. We rephrased the corresponding section in the discussion at line 400 to 409:

*To estimate the actual $\Delta DIC/\Delta SO_4^{2-}$ ratio due to diagenetic processes, the slope of the correlation between produced DIC ($\Delta DIC$) and consumed sulfates ($\Delta SO_4^{2-}$) in the pore waters (Fig 10) has to be corrected for molecular diffusion following the equation proposed by Berner (1980). Accordingly, we used the diffusion coefficients determined by Li and Gregory (1973). Below 10 cm depth, the observed diffusion corrected $\Delta DIC/\Delta SO_4^{2-}$ ratio equals 1.8 ± 0.02. The deviation of this measured value, from the theoretical value of 2 can be linked to higher oxidation states of organic matter which increases the $SO_4^{2-}$ requirement for DIC production (in an extreme case, if methane undergoes oxidation, the $\Delta DIC/\Delta SO_4^{2-}$ ratio equals 1), carbonate precipitation lowering DIC concentrations or methanogenesis that increases DIC without consuming $SO_4^{2-}$ (Burdige and Komada, 2011; Antler et al., 2014).*

*and further at line 417:*

*Despite all these divers reactions that affect the $\Delta DIC/\Delta SO_4^{2-}$ ratio, they are balanced in a way that $\Delta DIC$ and $\Delta SO_4^{2-}$ correlate well and do not show a deviation in the slope throughout the whole sediment depth investigated (Figure 10).*

(473) I don't see any direct evidence in the paper that terrestrial organic matter is what is being degraded. It might be, but without evidence to support this I would not be so definitive.

The sentence in question was rephrased:
*This confirms that the biogeochemistry in the prodelta region is driven by the import and processing of material from the Rhône River (Cathalot et al., 2010, 2013).*

Furthermore, different studies showed, that the majority of the sediment fraction in the proximal domain is land derived. This fraction decreases in offshore direction. During the DICASE cruise, porewater was sampled for analysis of $\delta^{13}C$ and $\Delta^{14}C$ signatures of porewater DIC in order to evaluate what OM fraction actually undergoes mineralization. The results point in the direction, that land

derived material is the DIC source in the pore waters close to the river mouth. An article to publish these results is on its way.

**Manuscript revision (doi:10.5194/bg-2016-212): Responses to the referees' comments:**

We thank the reviewer for his helpful and constructive comments. In the following document, we answer the questions (in black) one by one in blue. Modifications that have been done in to the manuscript are written in *italics*.

**Reviewer #2**

All minor comments of Reviewer #2 were taken into account.

Rassmann and collaborators present a very nice dataset of sediment properties in the Rhône river delta. Based on direct (microelectrodes) measurements of pH and O2 and on pore-water analyses of DIC, TA, Ca2+ and SO42- along a gradient from the river mouth to the open Mediterranean Sea (3 domains considered), this manuscript aims to describe and understand the main diagenetic reactions that control these sediment properties and the impact of the sediment on the bottom water carbonate chemistry. I would recommend publication of this manuscript following the proposed minor modifications and an extensive copy-edition by a native speaker.

The manuscript was copy-edited by a native English-speaker (Patrick Laceby, LSCE)

Table 2 and Section 2.7 should not be presented in the Material and Methods section but more likely in the Discussion.

We moved Section 2.7 to the beginning of the discussion and rephrased it.

I do not believe CO2 dissolution should be presented as a diagenetic reaction.
Table 2 should be made much clearer and for instance updated by: 1) providing the full name of the presented reactions, 2) dividing into 3 parts with reactions occurring in the presence of oxygen (oxic mineralization and reoxidation of reduced species), in the anoxic section (anaerobic mineralization and precipitation of reduced species), and both: CaCO3 dissolution or precipitation. Reaction R2, it is not clear whether you present CaCO3 dissolution or precipitation or both. I would consider 2 lines, one for precipitation, and one for dissolution as their effects on TA, DIC, pH and Omega are opposite. Finally, I would move this reaction to the end of the table (see above).

We sub-divided Table 2 into the 3 suggested categories and split reaction 2 into two lines for dissolution and precipitation.

Why don't you show nitrate reduction? This can be in some cases an important pathway.

Waters of the Northwest Mediterranean Sea show low nitrate concentrations. Only a minor part of OM is mineralized by this pathway in the study area. Nevertheless, denitrification has been added to the reaction table. To justify not to discuss nitrate reduction in this area, we cite (Pastor, L., Cathalot, C., Deflandre, B., Viollier, E., Soetaert, K., Meysmann, F.J.R., Ulses, C., Metzger, E. and Rabouille, C.: Modeling biogeochemical processes in sediments from the Rhône River prodelta area (NW Mediterranean Sea), Biogeosciences, 8, 1351-1366, 2011a.) at line 387:

*In contrast to other nearshore environments, nitrate reduction has been shown to account only for 2-5*

*% of OM mineralization in the sediments of the prodelta of the Rhône whereas other anaerobic mineralization processes account for 30-40 % in the distal domain and up to 90 % in the proximal domain (Pastor et al., 2011a). Nitrate reduction produces less TA than DIC (TA/DIC ratio = 0.8/1) and thus lowers Ω.*

All reactions should be presented considering the same amount of OM mineralized (in some cases you have 1 or 2 moles of CH2O mineralized).

The stoichiometric coefficients in table 2 have been adjusted to 1 mol of $CH_2O$.

Figure 6 should be updated. First, you don't show the same Y-axis scale than on Figure 5, why is that? and furthermore, this scale in not the same between the 3 domains in this Figure 7.

We adjusted the scale on all figures to 40 cm depth.

I am a bit surprised by the very high heterogeneity that you found between stations in the Proximal domain and believe there are a number of mistakes to correct. For instance, you have DIC and TA data for station Z' until 30cm while you calculate pH up to 25. For station A', you seem to have DIC/TA data down to 25 cm and you calculate pH data down to 35 cm or more. Please check.

In effect, the area is highly heterogeneous, differences of 10 mmol/L in DIC or TA pore water concentrations at a same station in a certain depth are definitely possible.
We checked the figures for consistency. Indeed, some data points had disappeared on the graphs and were re-introduced.

Furthermore, I have calculated pH for station Z at the last sampled depth (between 20 and 25 cm) considering TA of 48 and DIC of 50 mmol/L, I end up with a pH of 7.18, far from the 7.8 shown in Figure 6. Again, this should be carefully checked.

We use CO2SYS with the thermodynamic constants from (Lueker, T. J., Dickson, A. G., Keeling, C. D.: Ocean $pCO_2$ calculated from dissolved inorganic carbon, alkalinity, and equations for $K_1$ and $K_2$ : validation based on laboratory measurements of $CO_2$ in gas and seawater at equilibrium, Mar. Chem., 70, 105-119, 2000.) with TP = 0.1 µmol/kg and TSi = 6.4 µmol/kg (Denis, L., Grenz, C.: Spatial variability in oxygen and nutrient fluxes at the sediment-water interface on the continental shelf in the Gulf of Lions (NW Mediterranean), Oceanologica Acta, 26, 373-389, 2003) and making the hypothesis that theses values are constant with depth (in lack of better data). Furthermore, we use the bottom water salinity of 37.5 and the in situ temperature of 16.0 °C. The water depth at this station is 18 m. The analysis has been done at a temperature of 25 °C.

For the data point cited:
Z(18cm): TA = 46.215 ± 0.474 mmol/kg, DIC = 45.440 ± 0.190 mmol/kg leads to  pH= 7.636 ± 0.059
Z(22cm): TA = 49.189 ± 0.504 mmol/kg, DIC = 47.363 ± 0.129  mmol/kg leads to  pH= 7.801 ± 0.048
and
Z(26cm): TA = 44.514 ± 0.484  mmol/kg, DIC= 44.506 ± 0.051 mmol/kg leads to pH=7.492  ± 0.051

If the discrepancies of your and our calculations persist, we should do a more detailed comparison of our methods with a whole dataset.

L284-287:

You should present the determination coefficients and the corresponding slopes for each domain separately.

As the slopes were not significantly different, we followed the suggestion of Reviewer #1 and reported the overall slope 1.01 with r² =  0.9982.

L291: I would really like to see a more detailed comparison between measured and calculated pH.

As already posted in the answer to Reviewer #1:

We added at line 292 :
*A linear relationship of the pH data measured with microelectrodes against calculated pH by CO2SYS shows a correlation with a slope of 1.01 +/- 0.02 and an r² = 0.7483 (graph not shown).*

We compared the pH values calculated from DIC and TA data with microelectrode data. As the porewaters represent an integration of a certain sediment zone, the average signal of the microelectrodes for the same zone was used. The size of the influenced zone was calculated following: Seeberg-Elverfeldt, J., Schlüter, M., Feseker, T., Kölling, M.: Rhizon sampling of porewaters near the sediment-water interface of aquatic systems; Limnology and Oceanography: Methods, 3, 361-371, 2005

[Figure]

Because we already include 10 figures, we did not include a figure showing the correlation.

Section 3.4. Why all stations are not shown for Ca2+ (D, A' and Z' missing), why 3 datasets for A ?

We do not have any data for stations D and Z'. In fact, there has been a confusion at station A: the data sets are from A and A' and a longer core at station A that has been removed in the reviewed version of the paper.

L43: This is not correct, following your Table 2, aerobic mineralization does not produce TA.

We rephrased:

*Aerobic and anaerobic reaction pathways contribute to the production of dissolved inorganic carbon (DIC), which creates acidification of the bottom waters. Anaerobic reactions lead as well to production of total alkalinity (TA)*

L152: I don' understand what is "the slope of the pH variation", please rephrase.

We rephrased:

*The calibration of the pH electrodes was carried out using NBS buffers, thus allowing the estimation of*

*the slope of the electrode signal in fonction of pH variation at onboard temperature. The slope was then recalculated at in situ temperature and the electrode signal variation was transformed into pH changes.*

L156: .. "using this BW value the micro electrode measured pH variations", please rephrase

We rephrased:

 *The pH of bottom waters was determined using the spectrophotometric method with m-cresol purple following Clayton and Byrne, (1993) and Dickson et al., (2007).  Pore water pH on the total proton scale (pH$_t$) was recalculated using the signal of the microelectrode adjusted to this BW value.*

L162: surface of 109 cm, this is not a surface.

The surface of the head of the moving unit of the lander, where the electrodes are mounted measures 109 cm²

L171: according to Broecker and Peng (1974) L180 to 191: please precise how many replicates were measured for each parameter (pH, O2, DIC and TA)

 *All bottom water concentrations were measured as triplicates. Small sample volumes in pore waters only allowed for replicates for the DIC, SO$_4^{2-}$ and Ca$^{2+}$ analysis.*

 L299: (Fig. 5) L313: > 95% calcite + >5 % Mg-calcite do not leave much room for aragonite: : :.

Yes, indeed, we did not detect any aragonite in this area. The drainage basin of the Rhône River is characterized by old carbonates and the calcifying organisms on site (foraminifers) produce calcite tests. There are no corals in this area.

In lack of better material, the aragonite standard used for the calibration of the X-ray analysis is natural coral powder, characterized to have less than 2 % of Ca content.  (Kindler, P., Reyss, J.-L., Cazala, C., Plagnes V.: Discovery of a composite reefal terrace of middle and late Pleisocene age in Great Inagua Island, Bahamas. Implications for regional tectonics and sea-level history, Sedimentary Geology, 194, 141-147, 2007). It's diffraction analysis gives the following diffractogramme with the clearly visible principal Ar peak at $2\theta = 26.297°$ and its secondary peak at 27.298° (both slightly shifted from the theoretical value) :

[Figure]

A typical diffractogramme of Rhone delta sediments looks like this (Station Z). We recognize clearly the principal calcite peak at 2θ = 29.468° with a little deformation at the right indicating the presence of magnesian calcite. The position of the aragonite peak is covered by the base of another peak at 2θ = 26.674. These diffractogrammes do not allow a better deconvolution of the peaks and we have an uncertainty about the material used to calibrate our DRX measurements. So the presence of aragonite cannot be completely excluded, but is unlikely.

[Figure]

Many grammatical and formatting errors in the Discussion.

Grammatical and formatting errors have been corrected.

All figures starting from Figure 2: Average OPD for each domain should be shown on these plots.

The vertical scale on figures 2 and 4 is in mm whereas the vertical scale on the following figures is given in cm. The average OPD for each domain is situated between the sediment water interface and the first scale trait, so it would be invisible on these figures.

Figure 2 legend: in situ in italics Figure 3 legend: what do the vertical error bars correspond to? Please add.

We added in the legend of the figure:

*Error bars are standard deviations between the diffusive fluxes calculated from the 5 single oxygen profiles measured at each station.*

Figures 5 and 7: I would use the same x-axis scale for all 3 domains

We decided not to use the same concentration scale because the gradients are very different. Adjusting the scale for all figures would hide the form of the profiles in the prodelta and distal domain. To alert the reader about difference between the three concentration scales, we added to the figure caption:

*For better visibility of the profiles in each domain, the scale of the concentrations has been individually adjusted for each domain.*

and for Figure 7
separate SO42- and Omega, consider using 2 figures. On Figure 7, please note that the legend box hides the axis, which should be avoided.

The figure has been split into individual figures for sulfate concentrations and calcium carbonate saturation states.

Figure 6 and 7, and legends: what do the horizontal error bars correspond to? Please add.

We added to the caption of figure 6:

[revised manuscript text omitted]

Figure 1

[Figure]

Figure 2

[Figure]

$$DOU = 3.49 + 18.47\exp(-dist/2.05)$$

Figure 3

[Figure]

Figure 4

665

[Figure]

Figure 5

[Figure]

Figure 6

675

680

[Figure]

690

Figure 7

[Figure]

Figure 8

[Figure]

Figure 9

695

700

[Figure]

Figure 10

**Tables**

Table 1: Stations investigated during the DICASE cruise in June 2014 with the main properties of bottom waters; dist = distance from the Rhône river mouth

| Station | Long. (°E) | Lat. (°N) | Dist. [km] | Depth [m] | T [°C] | Salinity | $O_2$ [µmol L$^{-1}$] | DIC [µmol L$^{-1}$] | TA [µmol L$^{-1}$] | $pH_t$ | $SO_4^{2-}$ [mmol L$^{-1}$] | $pCO_2$ (calculated) [µatm] |
|---|---|---|---|---|---|---|---|---|---|---|---|---|
| Z, Z' | 4.865 | 43.317 | 2.2 | 18.0 | 16.0 | 37.5 | 244.0 ± 0.3 | 2330 ± 1 | 2648 ± 3 | 8.118 ± 0.003 | 28.4 ± 0.3 | 364.1 |
| A, A' | 4.851 | 43.312 | 2.1 | 18.3 | 16.8 | 37.7 | 245.1 ± 0.3 | 2323 ± 4 | 2613 ± 17 | 8.072 ± 0.004 | 28.2 ± 0.4 | 407.3 |
| AK | 4.856 | 43.307 | 2.8 | 48.1 | 15.8 | 37.4 | 240.8 ± 0.1 | 2335 ± 4 | 2623 ± 3 | 8.085 ± 0.011 | 29.7 ± 0.3 | 393.6 |
| B | 4.818 | 43.295 | 3.0 | 66.2 | 15.0 | 37.7 | 213.2 ± 0.8 | 2372 ± 5 | 2628 ± 2 | 8.039 ± 0.015 | 28.7 ± 0.3 | 446.1 |
| K | 4.856 | 43.302 | 3.3 | 60.5 | 14.9 | 37.7 | 226.4 ± 0.2 | 2351 ± 5 | 2538 ± 5 | 7.916 ± 0.002 | 29.1 ± 0.3 | 596.6 |
| L | 4.885 | 43.304 | 4.4 | 58.2 | 15.2 | 37.6 | 230.9 ± 0.6 | 2340 ± 2 | 2612 ± 5 | 8.066 ± 0.002 | 29.7 ± 0.3 | 412.4 |
| C | 4.773 | 43.271 | 8.8 | 75.0 | 14.4 | 37.7 | 225.6 ± 0.4 | 2354 ± 2 | 2621 ± 10 | 8.067 ± 0.004 | 29.0 ± 0.3 | 411.5 |
| D | 4.738 | 43.256 | 12.8 | 80.0 | 14.9 | 37.6 | 214.5 ± 0.5 | 2388 ± 8 | 2605 ± 3 | 7.970 ± | 30.2 ± 0.3 | 531.3 |

| | | | | | | | | | 0.002 | | |
|---|---|---|---|---|---|---|---|---|---|---|---|
| E | 4.685 | 43.219 | 17.9 | 77.3 | 14.3 | 37.7 | 226.3 ± 0.3 | 2391 ± 6 | 2594 ± 5 | 7.952 ± 0.004 | 30.4 ± 0.3 | 553.3 |
| F | 4.649 | 43.164 | 24.3 | 77.0 | 14.8 | 37.7 | 230.2 ± 0.1 | 2364 ± 4 | 2600 ± 24 | 8.008 ± 0.006 | 30.3 ± 0.3 | 478.4 |

710

Table 2: Diagenetic reactions and their effect on the carbonate system (TA, DIC, pH and $\Omega$)

| | | Reaction | ΔTA/ΔDIC | ΔpH | Δ $\Omega$ |
|---|---|---|---|---|---|
| | **Carbonate chemistry** | | | | |
| R1 | $CO_2$ dissolution | $CO_2+H_2O \leftrightarrow H_2CO_3 \leftrightarrow HCO_3^- +H^+ \leftrightarrow CO_3^{2-}+2H^+$ | | - | - |
| R2a | Carbonate dissolution | $CaCO_3+H_2O+CO_2 \rightarrow Ca^{2+}+2HCO_3^-$ | + 2/1 | + | + |
| R2b | Carbonate precipitation | $Ca^{2+}+2\,HCO_3^- \rightarrow CaCO_3+H_2O+CO_2$ | -2/-1 | - | - |
| | **Aerobic reactions** | | | | |
| R3 | Aerobic mineralization | $CH_2O+O_2 \rightarrow HCO_3^- +H^+$ | 0/1 | - | - |
| R4 | Nitrification | $NH_4^+ +2O_2 \rightarrow NO_3^- +H_2O+2H^+$ | -2/0 | - | - |

| | | | | | |
|---|---|---|---|---|---|
| R5 | Iron oxidation | $4Fe^{2+}+O_2+10H_2O \rightarrow 4Fe(OH)_3+8H^+$ | -8/0 | - | - |
| R6 | Manganese oxidation | $2Mn^{2+} + O_2 + 4HCO_3^- \rightarrow 2MnO_2 + 4CO_2 + 2H_2O$ | -4/0 | - | - |
| | **Anaerobic Reactions** | | | | |
| R7 | Nitrate reduction | $CH_2O+0.8\,NO_3^-+0.8\,H^+ \rightarrow CO_2+0.4\,N_2+1.4\,H_2O$ | 0.8/1 | - | - |
| R8 | Manganese reduction | $CH_2O+2MnO_2+3H^+ \rightarrow HCO_3^-+2Mn^{2+}+2H_2O$ | 4/1 | + | + |
| R9 | Iron reduction | $CH_2O+4Fe(OH)_3+7H^+ \rightarrow HCO_3^-+4Fe^{2+}+10H_2O$ | 8/1 | + | + |
| R10 | Sulfate reduction | $CH_2O+\frac{1}{2}SO_4^{2-} \rightarrow HCO_3^-+\frac{1}{2}HS^-+\frac{1}{2}H^+$ | 1/1 | - | |
| R11 | FeS precipitation | $Fe^{2+}+HS^- \rightarrow FeS+H^+$ | -2/0 | - | - |
| R12 | FeS precipitation with sulfate recycling | $8Fe(OH)_3+9HS^-+7H^+ \rightarrow 8FeS+SO_4^{2-}+20H_2O$ | -2/0 | + | + |
| R14 | Pyrite precipitation | $8Fe(OH)_3+15HS^-+SO_4^{2-}+17H^+ \rightarrow 8FeS_2+28H_2O$ | 2/0 | + | + |
| R15 | Anaerobic methane oxidation | $CH_4+SO_4^{2-} \rightarrow HS^-+HCO_3^-+H_2O$ | 2/1 | + (if pH < 7.9) | + |
| R16 | Methanogenesis | $CH_2O \rightarrow \frac{1}{2}CH_4+\frac{1}{2}CO_2$ | 0/0.5 | - | - |

715